# NI Sampling: Accelerating Discrete Diffusion Sampling by Token Order Optimization

**Enshu Liu,**[*]  **Xuefei Ning,**  **Yu Wang,**
Department of EE, Tsinghua University
`les23@mails.tsinghua.edu.cn`
`foxdoraame@gmail.com`
`yu-wang@mail.tsinghua.edu.cn`

**Zinan Lin**[†]
Microsoft Research
`zinanlin@microsoft.com`

## Abstract

Discrete diffusion language models (dLLMs) have recently emerged as a promising alternative to traditional autoregressive approaches, offering the flexibility to generate tokens in arbitrary orders and the potential of parallel decoding. However, existing heuristic sampling strategies remain inefficient: they choose only a small part of tokens to sample at each step, leaving substantial room for improvement. In this work, we study the problem of token sampling order optimization and demonstrate its significant potential for acceleration. Specifically, we find that fully leveraging correct predictions at each step can reduce the number of sampling iterations by an order of magnitude without compromising accuracy. Based on this, we propose **Neural Indicator Sampling** (**NI Sampling**), a general sampling order optimization framework that utilize a neural indicator to decide which tokens should be sampled at each step. We further propose a novel trajectory-preserving objective to train the indicator. Experiments on LLaDA and Dream models across multiple benchmarks show that our method achieves up to $14.3\times$ acceleration over full-step sampling with negligible performance drop, and consistently outperforms confidence threshold sampling in the accuracy–step trade-off. Code is available at https://github.com/imagination-research/NI-Sampling.

## 1 Introduction

Diffusion-based large language models (dLLMs) (Austin et al., 2021; Lou et al., 2023; Sahoo et al., 2024) are a recently emerged paradigm of text generative modeling, which have attracted increasing attention (Nie et al., 2024; Hoogeboom et al., 2021; He et al., 2022; Reid et al., 2022; Sun et al., 2022; Nie et al., 2025; Ye et al., 2025; Wu et al., 2025). Currently, dLLMs have demonstrated pratical impact in large-scale systems like Mercury (Khanna et al., 2025), Gemini Diffusion (DeepMind), and Seed Diffusion (Song et al., 2025), highlighting their potential for both deployment and methodological innovation. Similar to continuous diffusion models (Sohl-Dickstein et al., 2015; Ho et al., 2020; Song et al., 2020) for image generation, which recover data from noise through a reverse denoising process, dLLMs start from a prior sequential discrete distribution and progressively transform it into the target distribution. Compared to traditional auto-regressive large language models (AR LLMs), dLLMs are not restricted to left-to-right decoding scheme and can flexibly choose the generation order of all tokens in the sequence. Additionally, dLLMs' ability of sampling multiple tokens per step suggests the potential to surpass AR LLMs in efficiency.

As mentioned above, the sampling order of tokens should be considered for dLLMs, i.e., how to determine which tokens to sample at each step. Since sampling multiple tokens simultaneously may break inter-token dependencies (Liu et al., 2024a), default samplers adopt the conservative strategy of generating only one token at each step (Nie et al., 2025; Ye et al., 2025; Kim et al., 2025),

---

[*]Work mostly done during Enshu Liu's internship at Microsoft Research
[†]Project advisor: Zinan Lin

denoted as *full-step sampling*. However, this leads to large number of sampling steps and results in significant inefficiency. Fast-dLLM (Wu et al., 2025) introduces a heuristic but effective method called *confidence threshold sampling*, where at each step all tokens whose predicted probabilities exceed a threshold $\epsilon$ are unmasked simultaneously. This strategy substantially reduces the number of sampling steps while enabling a trade-off between efficiency and accuracy by varying threshold.

In this paper, we observe that the efficiency of existing heuristic sampling strategies for dLLMs are sub-optimal and still leave substantial room for improvement. Specifically, existing empirical methods typically unmask a token only when its predicted confidence is large enough (Wu et al., 2025; Nie et al., 2025; Kim et al., 2025; Ye et al., 2025). However, we find that at each step, the model is actually capable of correctly predicting a large number of tokens, as the token with the highest probability aligns with the final generated token. Existing methods can reveal only a small subset of these tokens, as the predicted confidence for most positions remains below the threshold, resulting in underutilization of the model's predictions at each step and, consequently, a substantially larger number of sampling steps. Intuitively, if we could identify and sample all correct tokens at each step, the total number of generation steps could be significantly reduced.

Based on this insight, we propose **Neural Indicator Sampling** (**NI Sampling**), a new and general framework to **optimize** the token sampling order in dLLMs. Our approach introduces a lightweight neural indicator, which determines whether the currently predicted token should be sampled for all masked positions in the form of a binary classification task. At each sampling step, all tokens judged as positive by the neural indicator are revealed, allowing the sufficient utilization of model predictions. To train this indicator, we propose to label the generated data in a *trajectory-preserving* way, which ensures that the optimal indicator can accurately maintain the original high-quality, albeit inefficient, generated trajectory with a much faster sampling speed. Note that this framework is not limited on our training strategy, leaving room for further exploration.

Our main contributions can be summarized as follows:

- In Sec. 3, we demonstrate that selecting an appropriate token sampling order has the potential to yield substantial acceleration. Specifically, we identify the phnomenon that the inference results of the dLLM at each step are not fully utilized. We show that if all masked positions labelled as positive according to our trajectory-preseving criterion are unmasked at every step, the model can achieve up to **24× faster** than default sampling and **more than 3×** faster than confidence threshold sampling (Wu et al., 2025), while perfectly preseving the performance of the default full-step method. This observation highlights the remarkable space for step compression and motivates us to optimize the sampling order.
- In Sec. 4, we introduce **NI Sampling**, a general framework for optimizing the token sampling order in dLLMs. Concretely, **NI Sampling** trains a lightweight indicator that makes token-wise binary decisions on whether each masked position should be revealed at every step. All predicted tokens judged as positive are sampled simultaneously. Following Sec. 3, we leverage the trajectory-preserving criterion to generate supervision signals to train this predictor. Importantly, the predictor is generic and trained once for different tasks.
- In Sec. 5, we apply **NI Sampling** on LLaDA-8B-Instruct (Nie et al., 2025), LLaDA-1.5 (Zhu et al., 2025), and Dream-7B-Base model (Ye et al., 2025) and evaluate on mathematical and code datasets. Compared with the full-step sampling baseline, **NI Sampling** achieves up to 14.3× speedup with only negligible performance degradation, outperforming confidence threshold sampling significantly. Additionally, our accuracy-step trade-off consistently dominates that of confidence threshold sampling across all settings. By combining with KV caching technique (Wu et al., 2025), we achieve up to 25.0× acceleration compared to full-step sampling.

## 2 Preliminary

### 2.1 Discrete Diffusion Models

Given a discrete random variable with a finite support $\mathcal{X} = \{1, \ldots, R\}$, discrete diffusion models gradually perturb its distribution $p_{data}$ into a prior distribution $p_T$ through a predefined forward differential equation $\frac{dp_t}{d_t} = Q_t p_t$, over the interval $t \in [0, T]$ (Campbell et al., 2022; Lou et al., 2023). To recover $p_{data}$, they solve the corresponding reverse differential equation, $\frac{dp_t}{dt} = \overline{Q}_t p_t$, from $T$ back to 0, where $\overline{Q}_t$ denotes the reverse diffusion matrices parameterized by a neural network.

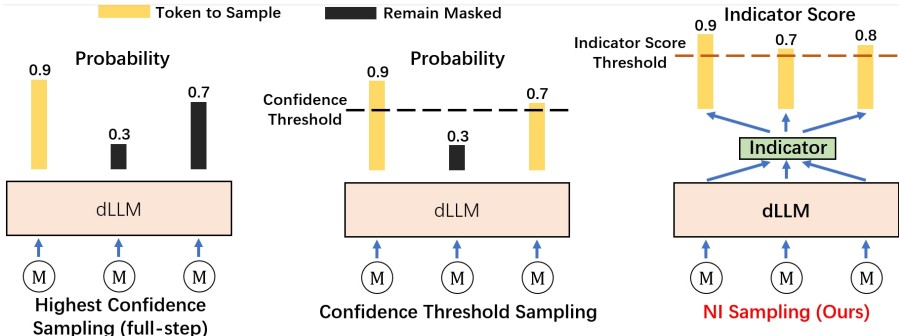

Figure 1: Comparison between **NI Sampling** and previous sampling methods. At each sampling step, **NI Sampling** uses a neural indicator to assign scores to masked positions. Tokens with sufficiently high indicator scores are then sampled.

**Masked Diffusion Models.** Among the various design choices for discrete diffusion model, setting $p_T$ as a delta distribution on the mask token $[MASK]$ and using $p_\theta(\boldsymbol{x}_0|\boldsymbol{x}_t)$ to parameterize $\overline{Q}_t$ emerges as the most widely adopted approach, known as masked diffusion models (MDMs). Specifically, MDMs treat the forward process as randomly masking tokens:

$$q(\boldsymbol{x}_t|\boldsymbol{x}_0) = \prod_{i=1}^{n} q(x_t^i|x_0^i) = \prod_{i=1}^{n} \mathrm{Cat}(x_t^i; (1-\alpha_t)\delta_{x_0^i} + \alpha_t \delta_{[MASK]}), \quad (1)$$

where $n$ is the sequence length, $t \in [0,1]$ denotes the diffusion timestep, and $\alpha_t$ specifies the noise schedule. MDMs then train the model to predict the conditional distribution of $\boldsymbol{x}_0$ given a noisy sequence $\boldsymbol{x}_t$, resulting in the following training loss:

$$\mathcal{L}(\theta) = -\mathbb{E}_{t,\boldsymbol{x_0},\boldsymbol{x_t}}(w_t \sum_{i=1}^{n} \mathbf{1}[x_t^i = [MASK]] \log p_\theta(x_0^i|\boldsymbol{x_t})), \quad (2)$$

where $x_0$ is sampled from the training set and $\mathbf{1}$ refers to the indicator function.

## 2.2 SAMPLING PROCESS OF MDMs

A well-trained MDM samples $\boldsymbol{x}_0$ by reversing the forward process Eq. (1), i.e., by iteratively unmasking from the wholely masked sequence $\boldsymbol{x}_1$. At each step, a set of masked positions is first selected to be revealed, and tokens are then sampled according to the predicted conditional probabilities at these positions. Given the prevalent use of greedy sampling in prior works (Ye et al., 2025; Nie et al., 2025; Zhu et al., 2025; Wu et al., 2025), which takes the token with highest probability as the sampling result, we adopt this setting in our paper.

Sampling multiple tokens will cause disalignment between the reverse and forward distributions. For example, for two masked positions $i_1$, $i_2$ at some step $t$, the predictions of the MDM at these positions $p_\theta(x_0^{i_1}|\boldsymbol{x}_t)$ and $p_\theta(x_0^{i_2}|\boldsymbol{x}_t)$, do not account for each other's sampling results, since both $x_t^{i_1}$ and $x_t^{i_2}$ are masked. Consequently, sampling them simultaneously would neglect correlations (Liu et al., 2024a; Wu et al., 2025). Therefore, typical methods restrict sampling to a single token at each step. Various strategies are proposed to select the masked token position $i^*$ to unmask at each step. **Top-1 probability** (Nie et al., 2025; Chang et al., 2022) selects the token with the highest probability: $i^* = \arg\max_{i \in \{i|x_t^i = [MASK]\}} \max_{j \in \{1,2,...,V\}} p_\theta(x_0^i = j|\boldsymbol{x}_t)$, where $V$ is the number of tokens in the codebook. **Top-1 probability margin** (Kim et al., 2025) selects the position with the largest gap between the top-1 and the top-2 probabilities: $i^* = \arg\max_{i \in \{i|x_t^i = [MASK]\}} |p_\theta(x_0^i = j_1|\boldsymbol{x}_t) - p_\theta(x_0^i = j_2|\boldsymbol{x}_t)|$, where $j_1$ and $j_2$ are tokens with top-1 and top-2 probabilities at position $i$, respectively. **Top-1 entropy** (Ye et al., 2025) selects the position with the highest entropy: $i^* = \arg\max_{i \in \{i|x_t^i = [MASK]\}} \sum_{j=1}^{V} p_\theta(x_0^i = j|\boldsymbol{x}_t) \log p_\theta(x_0^i = j|\boldsymbol{x}_t)$

The aforementioned methods suffer from slow generation due to the large number of steps. **Confidence threshold sampling** (Wu et al., 2025) is proposed to address this issue. At timestep $t$, it samples all tokens with probability exceeds a threshold $\epsilon$: $\{i| \max_j p_\theta(x_0^i = j|\boldsymbol{x}_t) \geq \epsilon\}$. It can be proved that with a sufficiently high threshold, the correspondence loss from sampling multiple tokens is negligible under greedy decoding (Wu et al., 2025). In practice, $\epsilon$ is typically set to 0.9.

| **Algorithm 1** Counting Mergeable Steps | **Algorithm 2** Merge along the trajectory |
|---|---|
| **Require:** | **Require:** |
|    A trajectory $\tau$ defined as Eq. (4); Step $k$ of the trajectory and $\boldsymbol{x}_k$; Pre-trained dLLM $\theta$. |    A reference trajectory $\tau$ defined as Eq. (4) with $n$ steps; Pre-trained dLLM $\theta$. |
| 1: $idx \leftarrow k + 1$ | 1: $step \leftarrow 1$, $\tau_{new} \leftarrow ()$ // initialize the new trajectory with an empty list |
| 2: **while** True **do** | 2: **while** $step \leq n$ **do** |
| 3:    **if** $\forall i \in A_{idx}, \mathrm{argmax}_{j \in \{1,...,V\}}\, p_\theta(x_0^i = j \vert \boldsymbol{x}_k) = x_*^i$ **then** | 3:    $idx \leftarrow$ run Alg. 1 with $\tau$,$step$ (as the $k$), and $\theta$. |
| 4:       $idx \leftarrow idx + 1$ | 4:    Add $A_{step} \cup \ldots \cup A_{idx-1}$ to $\tau_{new}$ |
| 5:    **else** | 5:    $step \leftarrow idx$ |
| 6:       **break** | 6: **end while** |
| 7:    **end if** | 7: **return** $\tau_{new}$ |
| 8: **end while** | |
| 9: **return** $idx$ | |

# 3 Revealing the Significant Acceleration Potential of Sampling Order Optimization

In this section, we demonstrate the substantial potential in reducing the number of sampling steps of optimizing the token sampling order. In Sec. 3.1, we formalize the optimization problem and space of selecting token sampling order. In Sec. 3.2, we show that selecting the sampling order in a trajectory-preserving manner can unlock significant acceleration ratio, thereby highlighting the value and importance of this optimization dimension.

## 3.1 Definition of Sampling Order Optimization

As discussed in Sec. 2, at each step we need to determine which positions to unmask. Since MDMs do not remask tokens that have already been sampled during the sampling process, the positions selected at different sampling steps do not overlap. Therefore, *sampling order optimization* can be formulated as an ordered partitioning of all token positions. Specifically, given a pre-trained dLLM $\theta$, a user-specified prompt $c$, and a generation length $N$, our goal is to find a sequence of sets $\boldsymbol{A} = (A_1, \ldots, A_n) = (\{i_{1,1}, \ldots, i_{1,a_1}\}, \ldots, \{i_{n,1}, \ldots, i_{n,a_n}\})$ that maximizes the reward:

$$\boldsymbol{A}^* = \underset{\boldsymbol{A} \in \mathrm{OPart}(S)}{\arg \max}\; R(\boldsymbol{A}, c, \theta), \tag{3}$$

where $S$ is the set $\{1, \ldots, N\}$; OPart refers to the ordered partition operation, giving $\mathrm{OPart}(S) = \{(A_1, \ldots, A_n) \vert \bigcup_{t=1}^{n} A_t = S, \forall t \quad A_t \neq \emptyset, \quad \forall t_1 \neq t_2 \quad A_{t_1} \cap A_{t_2} = \emptyset\}$; $R(\cdot, c, \theta)$ denotes a reward function defined over ordered partitions of $S$, which depends on model parameters $\theta$ and user prompt $c$. In general, there is no limitation to the choice of $R$, such as the performance on a specific downstream task. In our scenario, we aim to minimize the number of sampling steps $n$ while maintaining the desired performance.

## 3.2 Trajectory Preserving Principle for Order Selection

Given a pre-trained model $\theta$ we can first apply an existing method (e.g., those described in Sec. 2.2) to generate a trajectory $\tau$ and a token order $(A_1, \ldots, A_n)$. We define the trajectory as follows:

$$\tau = (\{(i_{1,1}, x_*^{i_{1,1}}), \ldots, (i_{1,a_1}, x_*^{i_{1,a_1}})\}, \ldots, \{(i_{n,1}, x_*^{i_{n,1}}), \ldots, (i_{n,a_n}, x_*^{i_{n,a_n}})\}), \tag{4}$$

where $\{i_{k,1}, \ldots, i_{k,a_k}\} = A_k$ is the set of indices of the selected positions, $x_*^{i_{k,m}} \in \{1, \ldots, V\}$ is the sampled token at position $i_{k,m}$.

We consider the $k$-th step of the trajectory, where the dLLM originally samples $a_k$ tokens at positions $i_{k,1}, \ldots, i_{k,a_k}$. At this point, consider the model's predictions at the selected positions in the next step $A_{k+1}$. If the model has already predicted all tokens at $A_{k+1}$ correctly, i.e.,

$$\forall i \in A_{k+1},\, \underset{j \in \{1,...,V\}}{\arg \max}\, p_\theta(x_0^i = j \vert \boldsymbol{x}_k) = x_*^i, \tag{5}$$

then merging the $k$-th and $(k+1)$-th steps does not change the following part of the trajectory. In other words, all tokens from the original $k$-th and $(k+1)$-th steps can be sampled simultaneously in only one step. For the step $k$, this merging process can be repeated iteratively until a new step $A_{k+s}$ does not meet the condition in Eq. (5), at which point the merging stops. The algorithm for merging a trajectory starting from step $k$ is presented in Alg. 1.

This merging operation can reduce the number of steps, but it requires the model to possess strong "jump-step" prediction capability. So, how well do current pre-trained dLLMs perform in this regard? We conducted a validation experiment to reveal their potential. We firstly obtain a reference

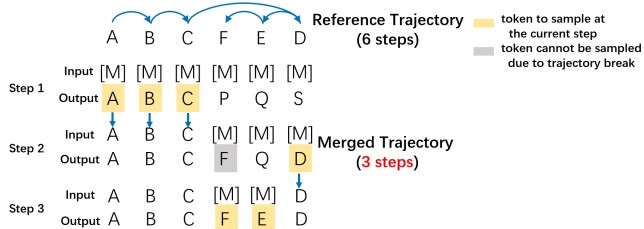

Figure 2: An example of **Trajectory-Preserving-Order**. At step 1, tokens A, B, and C are predicted correctly and follow the order of the reference trajectory, so they can be sampled. At step 2, token D, which is the next token in the reference trajectory, can be sampled. Although token F is predicted correctly, it cannot be sampled because its preceding token in the reference trajectory (E) is incorrect. At step 3, token E and F are predicted correctly and can be sampled.

Table 1: Performance of Trajectory-Preserving-Order.

| Model | Method | GSM8K-256 | | MATH-256 | | MBPP-256 | |
|---|---|---|---|---|---|---|---|
| | | Acc | Step | Acc | Step | Acc | Step |
| LLaDA-8B-Instruct | Full-step | 77.63 | 256 | 31.89 | 256 | 36.60 | 256 |
| | Threshold | 77.33 | 74.34 (3.4×) | 31.68 | 95.98 (2.7×) | 37.60 | 33.48 (7.6×) |
| | Traj-Preserving | 77.63 | 22.36 (11.4×) | 31.89 | 27.72 (9.2×) | 36.60 | 10.52 (24.3×) |
| LLaDA-1.5 | Full-step | 81.05 | 256 | 33.18 | 256 | 38.40 | 256 |
| | Threshold | 81.58 | 72.92 (3.5×) | 33.18 | 95.41 (2.7×) | 38.40 | 41.89 (6.1×) |
| | Traj-Preserving | 81.05 | 22.31 (11.5×) | 33.18 | 28.39 (9.0×) | 38.40 | 11.28 (22.7×) |

trajectory generated with the top-1 probability method with full-step (Chang et al., 2022; Nie et al., 2025). Starting from the first step of this trajectory, we iteratively perform step merging according to Alg. 1. When a merging round terminates due to a mismatch between the predicted tokens and the reference tokens, the next merging round begins from the step where the mismatch occurred. The loop continues until all steps in the original trajectory have been merged. We denote this strategy as **Trajectory-Preserving-Order**, with the full algorithm presented in Alg. 2 and Fig. 2.

Then, we validate this approach with LLaDA-8B-Instruct and LLaDA-1.5 models on GSM8K, MATh, and MBPP with generation length 256. The average numbers of steps required for the new trajectory, along with the performance, are shown in Tab. 1. Compared with confidence threshold sampling, **Trajectory-Preserving-Order** achieves more than 3× acceleration. When compared with the original top-1 probability trajectory with full-step, it can accelerate up to 24.3×. Moreover, this ordering strategy clearly preserves all sampling results from the reference trajectory, while confidence threshold sampling may incur performance degradation.

It is worth noting that Trajectory-Preserving-Order is only one of the possible strategies. More aggressive strategies may yield even greater improvements (e.g., 36.8× speedup. See App. B for details). We mainly discuss Trajectory-Preserving-Order because it strictly guarantees consistency between the new outputs and the reference trajectory. While this constraint limits its potential for achieving larger speedups, the acceleration it provides is already highly attractive.

Although these methods cannot be applied directly due to the absence of a reference trajectory, the results still demonstrate a promising space for sampling order optimization, which we study next.

# 4 NI SAMPLING: A GENERAL FRAMEWORK FOR SAMPLING ORDER OPTIMZIATION

In this section, we introduce **Neural Indicator Sampling** (**NI Sampling**), a general framework designed to address the optimization problem defined in Eq. (3). As discussed in Sec. 4.1, the key idea is to employ a neural indicator that evaluates every masked position and determines whether it should be sampled at each step. We further explain the training procedure in Sec. 4.2, and describe its input features and the model architecture of the neural indicator in Sec. 4.3.

## 4.1 A LIGHT NEURAL NETWORK AS TOKEN-WISE SAMPLING INDICATOR

We view Eq. (3) as the problem of determining which positions should be selected for sampling at each step. To this end, we propose a **token-wise neural indicator**, denoted as $\phi$, to make decisions. Specifically, after the inference of the dLLM, suppose there are $M$ masked positions remaining. The neural indicator treats the decision of whether to reveal each masked position as a binary classifi-

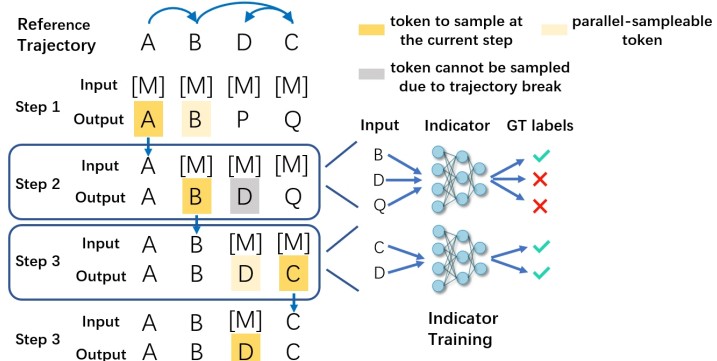

Figure 3: Generating training data for the indicator. At each step of the reference trajectory, labels for all tokens are assigned according to Alg. 1. For example, at step 3, token C, which is generated in the reference trajectory, is labeled positive; since the next token D is also predicted correctly, it is labeled positive as well. At step 2, token B, which is originally generated by the reference trajectory at this step, is labeled positive. The predicted token Q is incorrect (the correct one should be C) and therefore receives a negative label. Although token D is predicted correctly, it is assigned a negative label because its preceding token in the reference trajectory is not predicted correctly at this step.

cation task, producing $M$ scores $s_1, \ldots, s_M$. These indicator scores can be viewed as a new type of confidence, serving as the criterion for decision-making. For instance, one can simply select the token with the highest score. Since an additional network is employed to process the current states, it offers greater flexibility than directly using the model outputs as confidence, and the latter can in fact be regarded as a special case of our method. For the goal of acceleration, we also introduce a threshold parameter $\epsilon_\phi$ like confidence threshold sampling (Wu et al., 2025), and reveal all positions $i$ such that $s_i \geq \epsilon_\phi$. This mechanism enables a natural trade-off between speed and accuracy. To guarantee that at least one token is sampled, we first apply an existing sampling strategy to select a subset of tokens, and then use the neural indicator to further select among the remaining positions. The complete procedure for sampling with **NI Sampling** is summarized in Alg. 4. Besides, **NI Sampling** is also compatible with random sampling, with details in App. C. Since the neural indicator is trained with additional supervision signals, it encodes richer information than the raw probability vector alone, and thus has the potential to yield better sampling decisions. Note that we constrain the parameter size of the indicator to make sure the additional computational cost is negligible, with detailed results shown in Sec. D.2.

## 4.2 TRAINING THE NEURAL INDICATOR WITH TRAJECTORY PRESERVING PRINCIPLE

In this section, we describe the training procedure for the neural indicator. Following Sec. 3.2, we adopt a trajectory-preserving criterion to construct supervisory signals for the predictor, as it ensures that the reference trajectory is fully preserved when trained to optimality, which makes the training more stable. Specifically, we first use the pretrained dLLM to generate a dataset with trajectories $\tau$. During training, we sample a trajectory from the dataset, denoted as $\tau_d = (\{(i_{1,1}, x_*^{i_{1,1}}), \ldots, (i_{1,a_1}, x_*^{i_{1,a_1}})\}, \ldots, \{(i_{n,1}, x_*^{i_{n,1}}), \ldots, (i_{n,a_n}, x_*^{i_{n,a_n}})\})$. Then we randomly mask it along the trajectory. Concretely, we randomly select an integer $t' \in 0, \ldots, n-1$, then replace all positions $i_{k,m}$ with $k > t'$ by the mask token, and denote the set of these positions as $\mathcal{M} = \{i_{k,m} \mid k > t', \forall m\}$. Starting from this step, we follow the principle of trajectory-preserving in Alg. 1 to determine whether subsequent steps can be merged. All positions in mergeable steps are assigned a label of 1, while the others are assigned 0. We then collect these labels along with the input features in all masked positions, and train the indicator with cross-entropy loss. The algorithm is summarized in Alg. 3. Note that the data generation process can be done with any sampling method, making **NI Sampling** compatitable with all existing samplers.

## 4.3 DESIGN DETAILS OF THE NEURAL INDICATOR

In this section, we provide details of the neural indicator, including its inputs and architecture.

**Inputs.** The model should have explicit access to at least two types of information. First, it should know the token that will be sampled at the position (i.e., the token with the highest probability under the greedy setting), since knowledge of the sampling result is necessary before deciding whether

to retain it. Second, the model should have sufficient contextual information, as the appropriate decision for the same token may vary depending on the surrounding context. Inspired by existing sampling methods, we find it beneficial to provide the indicator with an explicit measure of the dLLM's confidence in its current prediction. This at least allows the neural indicator to reproduce existing methods. Moreover, beyond the top-1 predicted token at each position, the probabilities of other tokens can also provide valuable semantic information. Accordingly, for each position, the predictor receives the following inputs from each masked positions:

- The embeddings of the top-$K_1$ probability tokens, since the top-1 token is the sampled result and top-2 to $K_1$ tokens provide additional information.
- The last-layer hidden states, which obtained rich global information across the sequence.
- The top-$K_2$ logits of the current output, to help the indicator assess the dLLM's confidence.

**Architecture.** For simplicity, we adopt a position-wise MLP architecture, where each masked position is processed independently without explicit interaction between positions. This is feasible because contextual information is already embedded in the inputs. Different types of input features are first processed by separate linear layers, then concatenated and fed into the backbone of the indicator. The backbone consists of several stacked blocks, each comprising two linear layers, an activation function, and residual connections. Finally, a projection head maps the backbone outputs to 2-dimensional logits, followed by a softmax to produce the output score $s_i$ for position $i$.

## 5 EXPERIMENTS

### 5.1 EXPERIMENTAL SETUP

**Base Model and Benchmark.** We mainly evaluate **NI Sampling** with LLaDA-8B-Instruct (Nie et al., 2025) and LLaDA-1.5 (Zhu et al., 2025). To further demonstrate the generalization ability of our method across different model families, we additionally apply it on Dream-7B-Base (Ye et al., 2025), a model fine-tuned from an AR LLM. For benchmarks, we follow Wu et al. (2025) and select GSM8K (5-shot), MATH (4-shot), HumanEval (0-shot), and MBPP (3-shot) as evaluation datasets. We also report results under varying generation lengths, including 128, 256, and 512 tokens.

**Baselines and Evaluation.** We report the speedup of our method compared to full-step sampling and use confidence threshold sampling as the baseline. To measure sampling efficiency, we report both the average number of steps and tokens per second. Note we have already taken the overhead of the neural indicator into account when calculating tokens per second. We further present the detailed overhead of the predictor in Sec. D.2, along with additional implementation details in Sec. D.5.

**Setup of NI Sampling.** We use user inputs from ShareGPT dataset as prompts and generate 204k trajectories under three different generation lengths (128, 256, and 512) to train the neural indicator. Then we test this neural indicator across all evaluation settings, demonstrating its generality across tasks and generation lengths. For the LLaDA family of models, we adopt confidence-threshold sampling with the threshold fixed at 0.8 to efficiently generate training data. For the Dream model, we find that using full-step trajectories yields better performance. Additional details in Sec. D.5.

### 5.2 MAIN RESULTS

We present the results of **NI Sampling** with the LLaDA family of models in Tab. 2. As shown in the table, compared to full-step sampling, **NI Sampling** achieves up to around 15× speedup, while the performance degradation of **NI Sampling** is negligible across all datasets. **NI Sampling** even show slightly better performance than full-step sampling on some datasets, which may be due to the variance in evaluation. When compared with confidence threshold sampling, **NI Sampling** consistently delivers higher speedups under all settings, reaching up to 2.2× faster than it (e.g., 219.3 v.s. 100.8 token/s), while also attaining superior performance under most settings. These results highlight the strong effectiveness of **NI Sampling**.

**Trade-off between Performance and Efficiency.** To more comprehensively compare **NI Sampling** with confidence threshold sampling, we construct trade-off curves between performance and number of sampling steps for both methods by varying the probability threshold and predictor threshold (Fig. 4). More results can be found in Sec. D.3. We see that our method Pareto-dominates confidence threshold sampling across all settings, further demonstrating its practicality under different computational constraints.

Table 2: Comparison of **NI Sampling**, full-step, and confidence threshold sampling methods.

| Dataset | Method | LLaDA-8B-Instruct | | | LLaDA 1.5 | | |
|---|---|---|---|---|---|---|---|
| | | Acc | Steps | Token/s | Acc | Steps | Token/s |
| GSM8K-128 | Full | 73.92 | 128 | 20.4 | 76.65 | 128 | 19.7 |
| | Threshold | 73.77 | 49.35 | 53.3 (2.6×) | 75.82 | 47.44 | 53.5 (2.7×) |
| | **NI Sampling** | 73.69 | 34.36 | 76.5 (3.8×) | 76.19 | 29.89 | 85.3 (4.3×) |
| GSM8K-256 | Full | 77.63 | 256 | 18.6 | 81.05 | 256 | 18.3 |
| | Threshold | 77.33 | 74.34 | 62.9 (3.4×) | 81.58 | 72.92 | 65.0 (3.6×) |
| | **NI Sampling** | 77.18 | 50.97 | 90.2 (4.9×) | 80.67 | 53.59 | 85.6 (4.7×) |
| GSM8K-512 | Full | 74.83 | 512 | 14.4 | 80.67 | 512 | 15.0 |
| | Threshold | 75.28 | 73.29 | 104.4 (7.2×) | 80.89 | 72.38 | 105.6 (7.0×) |
| | **NI Sampling** | 76.57 | 51.08 | 147.0 (10.2×) | 81.20 | 53.56 | 140.6 (9.4×) |
| MATH-128 | Full | 29.64 | 128 | 28.5 | 31.03 | 128 | 27.8 |
| | Threshold | 29.69 | 60.43 | 57.9 (2.0×) | 30.93 | 58.71 | 60.9 (2.2×) |
| | **NI Sampling** | 30.07 | 39.01 | 87.2 (3.1×) | 31.41 | 39.65 | 87.1 (3.1×) |
| MATH-256 | Full | 31.89 | 256 | 25.0 | 33.18 | 256 | 25.0 |
| | Threshold | 31.68 | 95.98 | 66.7 (2.7×) | 33.18 | 95.41 | 67.0 (2.7×) |
| | **NI Sampling** | 31.67 | 61.92 | 98.7 (4.0×) | 32.98 | 69.67 | 89.5 (3.6×) |
| HumanEval-256 | Full | 37.80 | 256 | 44.1 | 43.90 | 256 | 44.7 |
| | Threshold | 37.20 | 98.66 | 115.8 (2.6×) | 42.68 | 100.7 | 113.3 (2.5×) |
| | **NI Sampling** | 37.20 | 55.60 | 192.5 (4.4×) | 42.68 | 64.40 | 168.4 (3.8×) |
| HumanEval-512 | Full | 35.37 | 512 | 31.5 | 40.85 | 512 | 32.1 |
| | Threshold | 35.37 | 158.6 | 100.8 (3.2×) | 39.02 | 158.4 | 101.1 (3.1×) |
| | **NI Sampling** | 35.98 | 69.54 | 219.3 (7.0×) | 39.63 | 105.3 | 144.2 (4.5×) |
| MBPP-256 | Full | 36.60 | 256 | 22.8 | 38.40 | 256 | 23.3 |
| | Threshold | 37.60 | 33.48 | 176.0 (7.7×) | 38.40 | 41.89 | 143.0 (6.1×) |
| | **NI Sampling** | 39.00 | 30.02 | 192.8 (8.5×) | 38.60 | 28.92 | 201.8 (8.7×) |
| MBPP-512 | Full | 36.80 | 512 | 19.1 | 37.80 | 512 | 18.7 |
| | Threshold | 37.60 | 38.75 | 250.2 (13.1×) | 38.60 | 44.38 | 215.8 (11.5×) |
| | **NI Sampling** | 38.00 | 35.25 | 263.9 (13.8×) | 38.80 | 34.62 | 268.0 (14.3×) |

Table 3: Results of **NI Sampling** combined with dual caching.

| Dataset | Method | Acc | Step | Speed |
|---|---|---|---|---|
| GSM8K-512 | Full-step | 74.83 | 512 | 14.4 |
| | **NI Sampling** | 75.44 | 44.85 | 197.7 (13.7×) |
| | NI Sampling+cache | 73.84 | 50.76 | 360.6 (25.0×) |
| HumanEval-512 | Full-step | 35.37 | 512 | 31.5 |
| | **NI Sampling** | 35.98 | 69.54 | 219.3 (7.0×) |
| | NI Sampling+cache | 35.98 | 94.39 | 247.3 (7.9×) |

Table 4: Results with Dream-7B-Base model.

| Dataset | Method | Acc | Steps | Token/s |
|---|---|---|---|---|
| GSM8K-256 | Full | 75.05 | 256 | 23.0 |
| | Threshold | 72.78 | 161.95 | 36.4 (1.6×) |
| | **NI Sampling** | 74.45 | 108.26 | 52.1 (2.3×) |
| MATH-256 | Full | 36.46 | 256 | 29.5 |
| | Threshold | 36.35 | 99.88 | 76.3 (2.6×) |
| | **NI Sampling** | 35.74 | 72.58 | 100.4 (3.4×) |
| MBPP-256 | Full | 57.60 | 256 | 29.7 |
| | Threshold | 53.40 | 92.41 | 82.6 (2.8×) |
| | **NI Sampling** | 56.00 | 68.87 | 106.7 (3.6×) |

**Combined with Caching.** We combine **NI Sampling** with Dual Cache method (Wu et al., 2025) to show that **NI Sampling** is compatible with other techniques for efficient dLLMs, with results shown in Tab. 3. With minimal performance loss, **NI Sampling** achieves more speedup (up to 25.0×).

## 5.3 ABLATION STUDY

**Base Model Type.** We test **NI Sampling**'s performance on a new base model Dream-7B-Base (Ye et al., 2025), with results shown in Tab. 4. Unlike LLaDA, which is a train-from-scratch dLLM, Dream is initialized from an AR LLM. Nonetheless, the results show that even with different base model training methods, **NI Sampling** still outperforms confidence threshold sampling.

**Training Set Distribution.** Our main experiments use the ShareGPT dataset to train a generic neural indicator. To examine the effect of training distribution, we train another neural indicator on a dataset combining the training sets of GSM8K and MATH. The results are shown in Fig. 5. We observe that the neural indicator trained on this mixed dataset performs better on the GSM8K and MATH test sets but worse on code datasets, likely because ShareGPT contains a higher proportion of code data. This suggests that neural indicator performance improves when the training distribution matches the test distribution more closely. Such a strategy can be employed to enhance **NI Sampling**'s performance for specific data domains, such as math-focused or code-focused tasks.

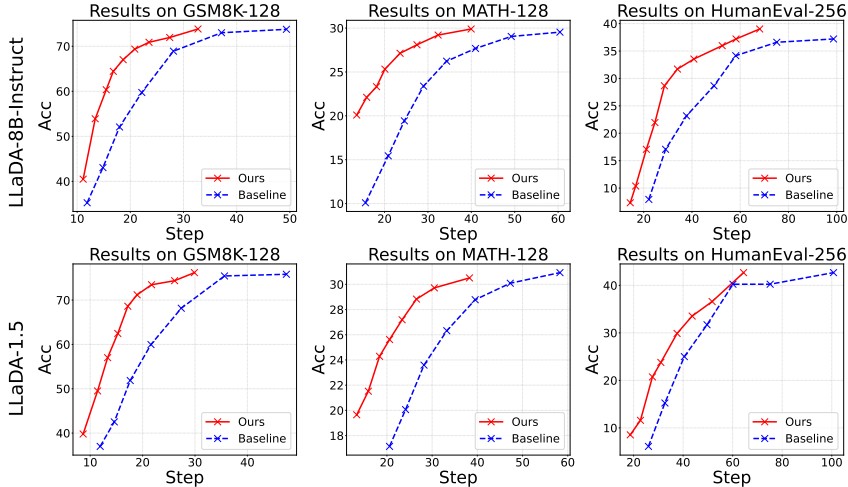

Figure 4: Trade-off curves between accuracy and steps with LLaDA models. More can be found in Sec. D.3.

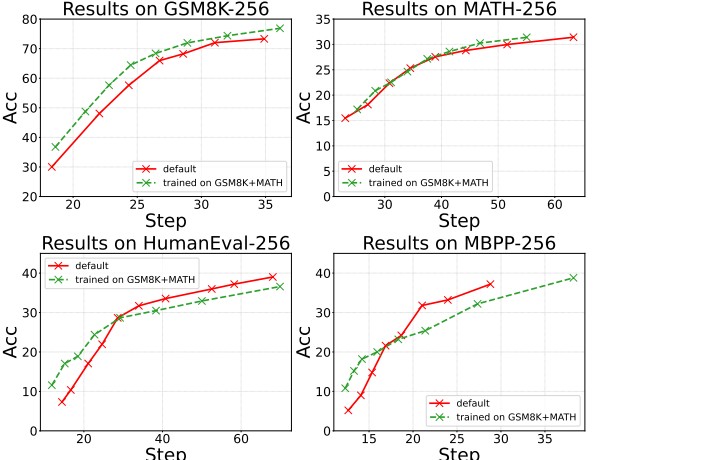

Figure 5: Ablation study of different training data distribution.

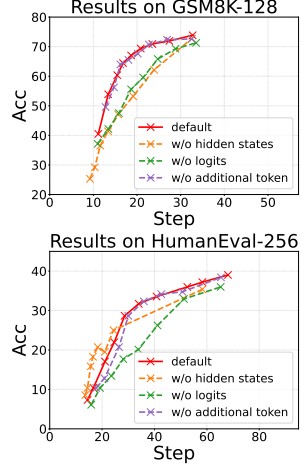

Figure 6: Ablation study on the input features.

**Input Types.** As discussed in Sec. 4.3, the neural indicator uses multiple types of inputs. We perform an ablation study on hidden states, predicted logits, and additional tokens to evaluate their contributions, with results shown in Fig. 6. It is evident that hidden states and predicted logits have a significant impact, and removing additional tokens also slightly degrades performance.

# 6 RELATED WORKS

## 6.1 CACHING FOR EFFICIENT DLLMS

In addition to reducing the number of sampling steps, caching has emerged as another promising approach for accelerating dLLMs. This strategy typically partitions the target sequence into blocks and caches the hidden states in a block-wise way. Wu et al. (2025) proposes caching either all non-current blocks or all prefix blocks, with the cache updated once the current block is generated. Liu et al. (2025b) introduces different caching strategies for prompts and responses. Arriola et al. (2025) further explored a semi-autoregressive architecture that caches all previously generated blocks while disregarding future blocks. It further conducts additional model training to support this design.

## 6.2 SAMPLING TECHNIQUES OF GENERATIVE MODELS

**Continuous Diffusion Models.** Continuous diffusion models view the generation as solving an reverse ODE. Prior works reduce the number of ODE steps by employing higher-order ODE solvers

(Lu et al., 2022; 2025; Zhao et al., 2023; Zhang & Chen, 2022). Other methods optimize sampling schedule to improve trade-offs between NFE and performance (Liu et al., 2024b; Zhou et al., 2024).

**Auto-regressive (AR) Models.** Speculative decoding is a widely adopted method for decreasing the number of AR models' forward passes (Xia et al., 2023; Cai et al., 2024; Li et al., 2024b;c; 2025; Teng et al., 2024), where a draft model generates multiple tokens that are then verified in parallel by the large AR model. Another line of works trains a token-level router that selectively assigns tokens to a large and a small AR model, thereby reducing the inference times of the large model. Liu et al. (2024a; 2025a) introduces flow matching and successfully reducing AR sampling to only one step.

**Image MDMs.** Predicting a set of tokens simultaneously and designing timestep schedule are used for accelerating image MDM sampling (Chang et al., 2022; Li et al., 2024a). Conceptually similar to this work, Token-Critic (Lezama et al., 2022a) and DPC (Lezama et al., 2022b) also apply an extra neural network to decide which token to remask at each step. However, they aim for better performance of image MDMs, which is different from our work.

## 7 LIMITATIONS AND FUTURE WORK

**Gap to the Upper Bound.** The speedup ratios reported in Sec. 5.2 are still far from those achieved by the analysis in Sec. 3.2 (e.g., 30.02 steps vs. 10.52 steps). This discrepancy arises because the indicator is not well trained. Improving the indicator's capability remains a promising direction.

**Potential Extensions.** In this work, we empirically adopt the trajectory-preserving principle as the optimization target for the indicator. In fact, many other objectives can be explored. For example, the final-results-preserving criterion described in App. B, although potentially less stable, provides a higher theoretical ceiling. Moreover, one could frame the indicator as an agent within a reinforcement learning paradigm, which takes actions over tokens at each step. This perspective would allow us to flexibly design reward functions for different targets, such as reasoning performance, and enable training the indicator within an RL framework.

## REPRODUCIBILITY STATEMENT

We describe the experimental details in Sec. 5.1 and Sec. D.5. The code will be open-sourced.

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

## A    ADDITIONAL ALGORITHM

We list the pseudo algorithm of sampling with **NI Sampling** and train the neural indicator with Trajectory-Preserving-Principle in this section.

---

**Algorithm 3** Train neural inidicator with Trajectory-Preserving-Principle

**Require:**
    Neural indicator $\phi$. Dataset $\mathcal{D}$ with trajectories; Pre-trained dLLM $\theta$.
1: **while** not converged **do**
2:    Sample $\tau$ from dataset $\mathcal{D}$.
3:    Sample $k$ from $\{1, \dots, n-1\}$.
4:    Get the $\boldsymbol{x}_k$ along $\tau$.
5:    $idx \leftarrow$ run Alg. 1 with $\tau$, $step$ (as $k$), and $\theta$.
6:    Label all indices that are within one of $A_k, \dots, A_{idx-1}$ as 1; The other indices are labled as 0.
7:    Collect the inputs and labels at all masked position and train the indicator within it.
8: **end while**
9: **return** $idx$

**Algorithm 4** Sample with the neural indicator

**Require:**
    A well trained neural indicator $\phi$; Pre-trained dLLM $\theta$; Indicator threshold $\epsilon_p$.
1: Initialize $\boldsymbol{x}$ with a sequence consisting only mask tokens.
2: **while** Sampling Unfinished **do**
3:    Calculate $p_\theta(\boldsymbol{x}_0|\boldsymbol{x})$.
4:    Reveal a subset of tokens with an existing sampling method.
5:    Collect all inputs at the masked positions and feed them into $\phi$ to get the indicator score.
6:    Reveal all tokens with indicator score larger than $\epsilon_p$, resulting in new $\boldsymbol{x}$.
7: **end while**
8: **return** $\boldsymbol{x}$

---

## B    FINAL-RESULTS-PRESERVING ORDER

In Sec. 3.2, we introduced **Trajectory-Preserving-Order**, a conservative approach in which a masked position is not necessarily revealed even if the model's current prediction at that position matches the final generation result. This is because doing so might affect final predictions at other masked positions if there exist intermediate positions between this position and the current step on the reference trajectory that are not aligned with the final result. Here, we relax this constraint: as long as the model's prediction at the current step aligns with the pre-obtained final generation result, then the token can be sampled at this step. We refer to this ordering strategy as Final-Results-Preserving Order. Using the top-1 probability full-step sampling results as the reference, we conducted experiments on GSM8K-256 with LLaDA-8B-Instruct and LLaDA 1.5 models. Results are reported in Tab. 5.

Table 5: Performance on GSM8K-256 of Final-Results-Preserving Order.

| Method | LLaDA-8B-Instruct | | LLaDA-1.5 | |
|---|---|---|---|---|
| | Acc | Steps | Acc | Steps |
| Full-step | 77.63 | 256 | 81.05 | 256 |
| Threshold | 77.33 | 74.34 (3.4×) | 81.58 | 72.92 (3.5×) |
| Trajectory-Preserving | 77.63 | 22.36 (11.4×) | 81.05 | 22.31 (11.5×) |
| Final-Results-Preserving | 77.78 | 6.95 (36.8×) | 80.89 | 9.11 (28.1×) |

Although Final-Results-Preserving-Order does not strictly guarantee identical generated results with the reference results, its performance remains nearly unchanged. We further examine the answers for the first 100 problems. For the LLaDA-8B-Instruct model, only 20 of the first 100 generated results produced by Final-Results-Preserving are not exactly identical to the reference results. In most cases, only a few tokens differ for these examples, while all final answers remain identical. For the LLaDA-1.5 model, 29 generations differ, among which only one produced a different answer. These observations indicate that this criterion causes negligible performance degradation while further increasing the acceleration ratio to a remarkably high level. Nevertheless, considering the training stability and task simplicity, we adopt the simpler yet efficient enough Trajectory-Preserving criterion as the training objective for the indicator.

## C  RANDOM SAMPLING WITH NI SAMPLING

As discussed in 4, the default sampling process of **NI Sampling** is deterministic. This does not align with the practical requirements of LLM applications, since users do not expect a single fixed output for the same prompt. In this section, we discuss the potential of **NI Sampling** for supporting random sampling. Specifically, we show how to introduce randomness with current indicator in Sec. C.1, which is trained by the process described in Sec. 4.2. Moreover, to prevent the indicator from relying too heavily on a deterministic trajectory, we propose to use random trajectory for training in Sec. C.2. Finally, we report the diversity evaluation results of **NI Sampling** in Sec. C.3.

### C.1  RANDOM SAMPLING WITH CURRENT INDICATOR

Our current indicator can be incorporated with random sampling easily without additional tuning. Specifically, similar to the implementation of full-step sampling and confidence threshold sampling, we sample token at each position randomly according to the probability vector instead of using argmax. Then, we feed all actually sampled tokens together with its probability into the existing neural indicator and let it make decisions. Other parts are the same as the description in Alg. 4.

We discuss the intuition behind the effectiveness of this design below.

**Our method follows the same paradigm as the random sampling implementation of confidence threshold sampling.** Specifically, under the greedy setting, both methods deterministically sample all tokens and keep the tokens whose scores exceed a threshold. When combined with random sampling, the modification remains exactly the same for both methods: deterministic selection is simply replaced by sampling from the predicted distribution. The only difference between the two methods lies in how eligible tokens are determined—confidence-threshold sampling uses confidence values, while our method uses the indicator score.

**In fact, our method may be even more diverse than confidence threshold sampling.** At each step, the set of tokens that can be accepted under greedy sampling is larger in our method than in confidence-threshold sampling. Consequently, under random sampling, there is a higher probability of selecting tokens that deviate from the original trajectory. This increases the likelihood of trajectory divergence and thus leads to greater sampling diversity.

### C.2  IMPROVING SAMPLING DIVERSITY BY TRAINING INDICATOR WITH RANDOM TRAJECTORY

The diversity of our method can be further improved by modifying the training process. Since our existing indicator has never seen data beyond tokens sampled by greedy sampling, nor trajectories other than deterministic ones, we revised the training strategy and retrained the model. The adjustments are:

- The deterministic trajectory generation process was replaced with a stochastic process using a temperature of 1.0. This can address the reviewer's concern regarding potential overfitting to deterministic trajectories.
- During training, tokens are sampled according to the model's predicted probabilities to serve as inputs of the indicator, rather than being selected solely with greedy strategy.

### C.3  RESULTS OF SAMPLING DIVERSITY

We evaluate the diversity of our random sampling algorithm in this section. We select LLaDA-8B-Instruct as the pre-trained model and set the temperature as 1.0 for all experiments. For each problem, we generate $k$ different answers using multiple random seeds, and report the following metrics to quantify diversity:

- **pass@k**. For this metric, a problem is considered correctly solved if at least one of the $k$ generated answers is correct. A diverse generative distribution should show a stable increase in pass@k as $k$ grows. Note that we use this metric for all datasets rather than being limited to code datasets. For SQuADv2, which uses average F1 score as the evaluation metric, we report the average of the maximum F1 scores across the $k$ independently generated

Table 6: Diversity evaluation results on HumanEval-256 dataset. "Slope" stands for the slope of the fitted line of pass@k with respect to $logk$.

|  | Token/s | NFE | Pass@1 | Pass@2 | Pass@4 | Pass@8 | Pass@16 | **Slope↑** |
|---|---|---|---|---|---|---|---|---|
| Full-step | 42.91 | 256 | 36.99 | 44.69 | 51.43 | 57.81 | 63.76 | 6.67 |
| Threshold | 110.82 | 99.13 | 36.82 | 44.35 | 51.02 | 57.40 | 63.10 | 6.56 |
| Old Indicator | 164.09 | 63.34 | 36.43 | 44.13 | 51.09 | 57.63 | 63.62 | 6.79 |
| New Indicator | 155.78 | 66.72 | 36.70 | 44.38 | 51.34 | 58.03 | 64.33 | **6.89** |

Table 7: Diversity evaluation results on GSM8K-256 dataset. "Slope" stands for the slope of the fitted line of pass@k with respect to $logk$.

|  | Token/s | NFE | Pass@1 | Pass@2 | Pass@4 | Pass@8 | **Slope↑** | 1-gram-BLEU↓ | 2-gram-BLEU↓ | MAUVE↓ |
|---|---|---|---|---|---|---|---|---|---|---|
| Full-step | 19.11 | 256 | 76.95 | 84.83 | 89.99 | 93.48 | 5.47 | 0.916 | 0.883 | 0.966 |
| Threshold | 65.56 | 74.63 | 77.55 | 85.51 | 91.05 | 93.85 | 5.44 | 0.920 | 0.887 | 0.974 |
| Old Indicator | 86.74 | 54.70 | 77.78 | 85.97 | 91.05 | 93.70 | 5.28 | 0.919 | 0.886 | 0.971 |
| New Indicator | 82.36 | 57.68 | 77.33 | 84.91 | 90.83 | 94.47 | **5.73** | 0.918 | 0.884 | 0.973 |

Table 8: Diversity evaluation results on Squad-Completion-128 dataset. "Slope" stands for the slope of the fitted line of pass@k with respect to $logk$.

|  | Token/s | NFE | Pass@1 | Pass@2 | Pass@4 | Pass@8 | **Slope↑** | 1-gram-BLEU↓ | 2-gram-BLEU↓ | MAUVE↓ |
|---|---|---|---|---|---|---|---|---|---|---|
| Full-step | 48.48 | 128 | 78.3 | 83.3 | 88.2 | 91.3 | 4.39 | 0.809 | 0.600 | 0.981 |
| Threshold | 174.64 | 34.03 | 77.5 | 84.0 | 87.0 | 89.5 | 3.90 | 0.802 | 0.600 | 0.986 |
| Old Indicator | 266.78 | 21.39 | 76.8 | 83.2 | 86.8 | 89.7 | 4.23 | 0.810 | 0.592 | 0.982 |
| New Indicator | 254.98 | 22.38 | 76.8 | 83.2 | 87.3 | 89.8 | **4.31** | 0.811 | 0.589 | 0.981 |

answers for all problems. We then report pass@k under different $k$ and further compute slope of the fitted line of pass@k with respect to $logk$, which serve as the primary metrics for evaluating diversity. Higher values indicate higher diversity.

- **Self-BLEU** (Zhu et al., 2018). For each answer, we compute its BLEU score with respect to the other $k-1$ answers. Then we take an average across all answers. We use both 1-gram-BLEU and 2-gram-BLEU. Lower values indicate higher diversity.
- **Self-MAUVE** (Pillutla et al., 2021). We randomly split the $k$ answers into two groups and compute the MAUVE score between them. Smaller values indicate higher diversity.

We demonstrate results in Tab. 6, Tab. 7, Tab. 8, Tab. 9, and Tab. 10. The row "Old Indicator" shows the diversity of our current neural indicator, while the row "New Indicator" shows the diversity of the newly trained indicator as discussed in Sec. C.2. The key takeaways are: **(1)** our existing indicator has already achieved comparable diversity to threshold sampling and full-step sampling across most datasets. The slope of pass@k with respect to $logk$ is slightly higher than that of confidence threshold sampling in most cases, and comparable to full-step sampling. This indicates that our method does not introduce additional loss of distributional diversity. Additionally, in most cases, our Self-BLEU and Self-MAUVE scores are comparable to or only slightly worse than those of confidence threshold sampling, further supporting the conclusion that there is no loss of distributional diversity in our method; **(2)** the row "New Indicator" shows a larger slope of pass@k with respect to $logk$, which indicates that training predictor on random trajectories can boost our sampling diversity.

## D  ADDITIONAL EXPERIMENTAL RESULTS AND DETAILS

### D.1  COMPARISON WITH MORE BASELINE

In this section, we compare **NI Sampling** with another related work, Token-Critic (Lezama et al., 2022a). As discussed in Sec. 6, Token-Critic also introduces an auxiliary neural network to decide

Table 9: Diversity evaluation results on Squadv2-128 dataset. All numbers of pass@k are F1 scores. "Slope" stands for the slope of the fitted line of pass@k with respect to $logk$.

|  | Token/s | NFE | Pass@1 | Pass@2 | Pass@4 | Pass@8 | **Slope↑** | 1-gram-BLEU↓ | 2-gram-BLEU↓ | MAUVE↓ |
|---|---|---|---|---|---|---|---|---|---|---|
| Full-step | 47.94 | 128 | 27.64 | 31.76 | 37.46 | 41.92 | 4.85 | 0.792 | 0.710 | 0.989 |
| Threshold | 202.27 | 31.23 | 28.05 | 33.21 | 37.92 | 42.21 | 4.72 | 0.792 | 0.710 | 0.989 |
| Old Indicator | 267.74 | 22.56 | 27.85 | 33.13 | 38.01 | 43.20 | 5.09 | 0.797 | 0.723 | 0.989 |
| New Indicator | 283.18 | 21.32 | 27.09 | 34.10 | 39.21 | 45.20 | **5.94** | 0.802 | 0.720 | 0.990 |

Table 10: Diversity evaluation results on TruthfulQA-128 dataset.

|  | Token/s | NFE | bleu_acc | rouge1_acc | rouge2_acc | rougeL_acc | 1-gram-BLEU↓ | 2-gram-BLEU↓ | MAUVE↓ |
|---|---|---|---|---|---|---|---|---|---|
| Full-step | 46.54 | 128 | 53.86 | 53.61 | 49.07 | 52.88 | 0.898 | 0.862 | 0.983 |
| Threshold | 218.80 | 26.46 | 55.32 | 54.35 | 49.20 | 54.10 | 0.903 | 0.867 | 0.984 |
| Old Indicator | 419.67 | 13.41 | 57.89 | 56.55 | 52.39 | 57.04 | 0.903 | 0.867 | 0.980 |
| New Indicator | 408.94 | 13.90 | 61.57 | 59.00 | 54.59 | 59.73 | 0.901 | 0.864 | 0.985 |

Table 11: Comparison between **NI Sampling** and Token-Critic (Lezama et al., 2022a).

| NI Sampling | Step | 15.47 | 20.84 | 27.41 |
|---|---|---|---|---|
|  | Acc | 60.35 | 69.37 | 71.95 |
| Token-Critic | Step | 15.08 (Top-8) | 22.01 (Top-4) | 28.58 (Top-2) |
|  | Acc | 33.59 | 42.84 | 53.83 |

Table 12: Comparison between indicator sizes of pre-trained dLLM and neural indicator.

|  | **Model Size** | **Indicator Size** |
|---|---|---|
| LLaDA | 8.01B | 84.2M |
| Dream | 7.62B | 96.4M |

which tokens should be unmasked during sampling. For a fair comparison, we adopt the same neural indicator architecture and training configuration for both our method and Token-Critic. Following the original paper, we apply top-k sampling to Token-Critic and combine it with confidence threshold sampling, consistent with NI sampling. Results on GSM8K are presented in Tab. 11. We observe that Token-Critic does not outperform NI sampling under accelerated sampling settings. One possible reason is that predicting whether a token is originally masked may not serve as an effective principle for speeding up autoregressive decoding.

## D.2 COST OF THE NEURAL INDICATOR

We first list the parameter size of the neural indicator and the pre-trained dLLM in Tab. 12. We further count the inference time of the neural indicator and the pre-trained dLLM, with results reported in Tab. 13 and Tab. 14. We can see that the indicator contains only about 1/80 to 1/100 of the parameters of the dLLM, while its inference time ranges from roughly 1/18 to 1/40. This clearly shows that the additional time and memory overhead introduced by **NI Sampling** is minimal. It is worth noting that we do not apply any system-level optimizations, so the indicator's time cost could be further reduced in practice. All the speed results we report in Sec. 5 also include the indicator's runtime.

## D.3 TRADE-OFF CURVES UNDER MORE SETTINGS

We present more performance-step trade-off curves, including LLaDA-8B-Instruct and LLaDA-1.5 on GSM8K-512, HumanEval-512 and MBPP-512 datasets, and Dream-7B-Instruct on GSM8K-256, MATH-256 and MBPP-256 datasets, in Fig. 7, Fig. 8 and Fig. 9, respectively. We also provide trade-off curves between accuracy and inference time, in Fig. 10, Fig. 11, Fig. 12 and Fig. 13.

Table 13: Inference time (ms) of LLaDA-8B-Instruct model and the neural indicator.

| Dataset | Model | Indicator |
|---|---|---|
| GSM8K-128 | 49.0 | 1.15 |
| GSM8K-256 | 53.8 | 1.56 |
| GSM8K-512 | 69.4 | 2.43 |
| MATH-128 | 35.1 | 1.12 |
| MATH-256 | 40.0 | 1.45 |
| HumanEval-256 | 22.7 | 1.13 |
| HumanEval-512 | 31.7 | 1.77 |
| MBPP-256 | 43.9 | 1.40 |
| MBPP-512 | 52.3 | 1.74 |

Table 14: Inference time (ms) of Dream-7B-Base model and the neural indicator.

| Dataset | Model | Indicator |
|---|---|---|
| GSM8K-256 | 43.4 | 1.82 |
| MATH-256 | 33.8 | 1.29 |
| MBPP-512 | 33.7 | 1.12 |

**NI Sampling** consistently pareto-dominate baseline method, demonstrating the effectiveness of our method.

### D.4 MORE ABLATION RESULTS

**Parameter Size of the Indicator.** We increase the parameter size of indicator by simply stacking more layers onto the MLP backbone, resulting in a larger indicator with 184M parameters. Fig. 14 presents a comparison between the two indicator sizes. It can be seen that increasing the indicator size does not provide obvious performance gains. For efficiency considerations, we therefore adopt the smaller indicator size in our main experiments.

**Training Set Size.** To reduce training cost, we decrease the amount of data used for training and evaluate its impact. The results are shown in Fig. 15, indicating that reducing the training data by 75% does not significantly hurt performance. Therefore, users can safely reduce the number of generated trajectories if faster training is needed.

### D.5 MORE EXPERIMENTAL DETAILS

#### D.5.1 EVALUATION

For the evaluation of performance, we follow Wu et al. (2025) and adopt lm-eval framework (https://github.com/EleutherAI/lm-evaluation-harness). We use the "flexible-extract" filter for GSM8K dataset, the average of "exact_match" and "math_verify" for MATH dataset, and pass@1 for both HumanEval and MBPP datasets. For sampling speed, we only count the GPU inference time of the pre-trained dLLM and the neural indicator for all settings to ensure a clean comparison. All inference speeds of LLaDA-8B-Instruct and LLaDA-1.5 are measured on a single NVIDIA H200 GPU, while those of Dream-7B-Base are measured on a single NVIDIA H800 GPU.

#### D.5.2 BASELINE

For full-step baseline, we follow the default setting in the official implementation of LLaDA and Dream. We use top-1 probability criterion for LLaDA-8B-Instruct and LLaDA-1.5 to choose the token sampled at each step. For Dream-7B-Base, we use the top-1 entropy for token selection. For confidence threshold sampling, we use 0.9 as the threshold for most cases reported in the Tab. 2. For several datasets, the accuracies of using 0.9 threshold are actually worse than a lower threshold, so we choose the best threshold for them. Specifically, for LLaDA-8B-Instruct model, we set the threshold as 0.7 on GSM8K-512, and 0.8 on MBPP 256; for LLaDA-1.5 model, we set the threshold

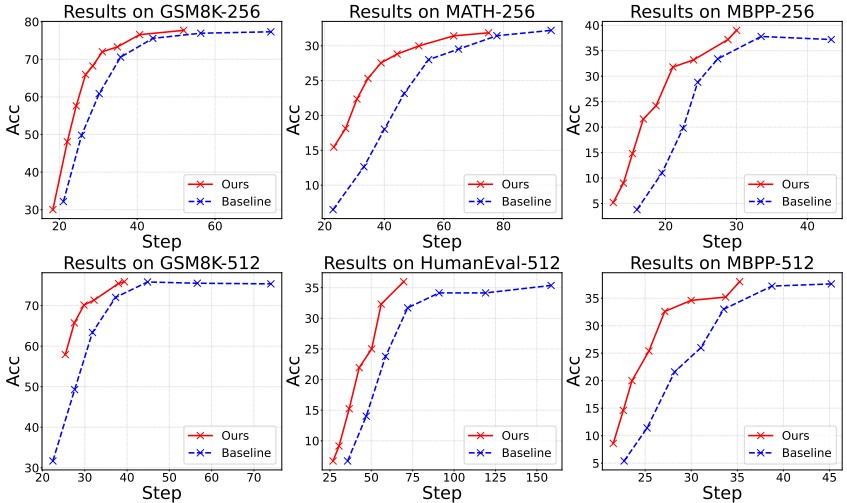

Figure 7: Performance-step trade-off curves of LLaDA-8B-Instruct.

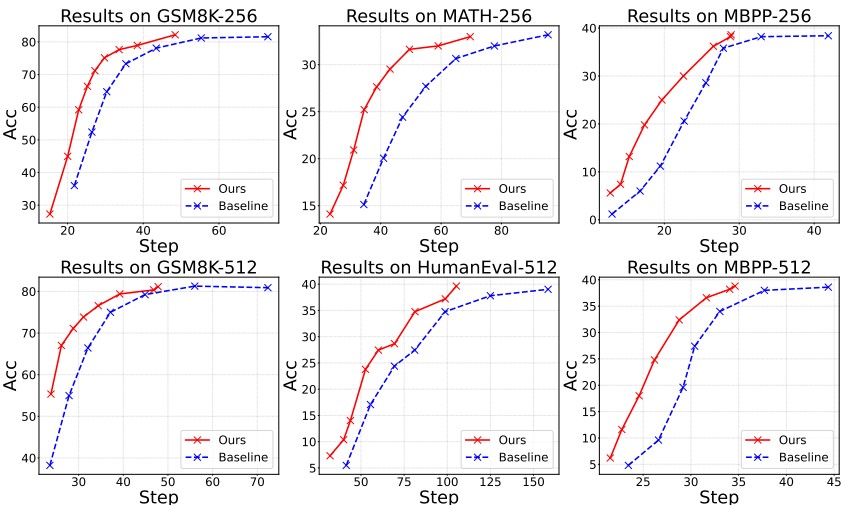

Figure 8: Performance-step trade-off curves of LLaDA-1.5.

as 0.8 on GSM8K-512. For trade-off curves, we choose threshold from 0.3 to 0.9 with an interval 0.1.

### D.5.3 NI SAMPLING

**Architecture of the Neural Indicator.** As discussed in Sec. 4.3, we use a MLP architecture for the neural indicator. We set the width of MLP as 768 and the depth of MLP as 5 for all main experiments with LLaDA-8B-Instruct and LLaDA-1.5 models. For Dream-7B-Base model, we change the depth of MLP to 8. For the ablation study in Fig. 14, we increase the layer number to 10 to construct a larger indicator.

**Data Generation.** In our main experiments, we use ShareGPT dataset to construct training set for the indicator. Following speculative decoding methods (Cai et al., 2024; Li et al., 2024b), we use the first prompt from the user and ignore other data. We do not conduct any additional operations and directly feed the prompt to the generation pipeline of dLLMs. For LLaDA-8B-Instruct and LLaDA-1.5 models, we generate each data sample three times, using generation lengths of 128, 256, and 512, respectively. We adopt confidence threshold sampling to construct reference trajectories and set the threshold as 0.8 for efficiency. For Dream-7B-Base model, we generate each dataset sample

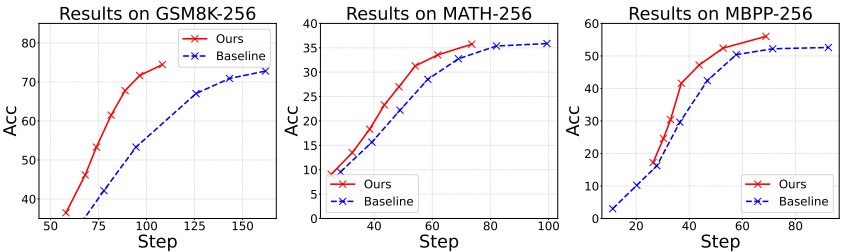

Figure 9: Performance-step trade-off curves of Dream-7B-Base.

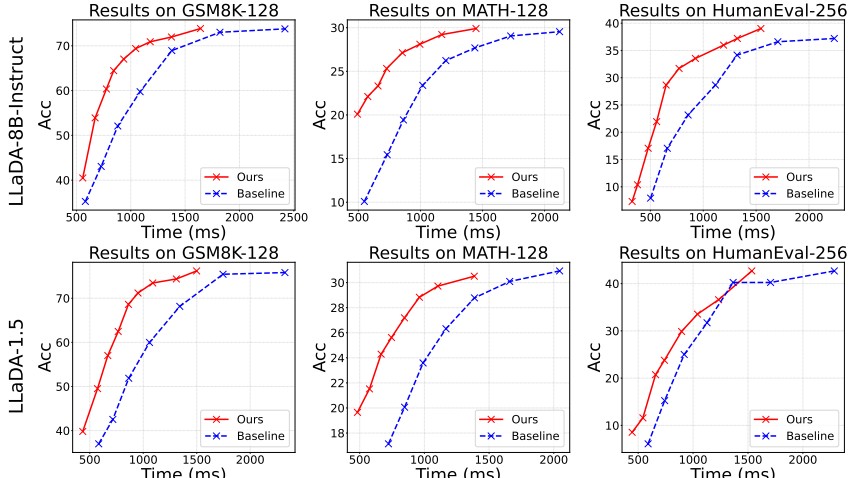

Figure 10: Performance-inference time trade-off curves of Dream-7B-Base.

twice with 128 and 256 generation lengths. We use the default top-1 entropy full-step sampling to generate reference trajectories, as we find that parallel generation is more challenging on Dream, possibly because Dream is initialized with an AR LLM.

**Training.** We train the neural indicator for 50 epochs under all settings. We apply AdamW optimizer, with a fixed learning rate of 0.0002 and betas of (0.9, 0.95), We set the batch size as 256. Empirically, to make the training task easier to learn, we additionally require that tokens labeled as positive must have a probability learger than 0.15; otherwise, they would be labeled as negative.

**Sampling.** As discussed in Sec. 4.1, we first sample a subset of tokens with an existing sampling method to prevent the situation where no token reaches the indicator score threshold. For most cases, we choose confidence threshold sampling with threshold 0.9 for results in Tab. 2 and threshold 0.8 for other data points in all trade-off curves on LLaDA models before we use indicator scores at each step. For those datasets mentioned in Sec. D.5.2 where 0.9 is not the optimal threshold, we used the corresponding optimal threshold instead. For Dream-7B-Base model, we consistently apply top-1 entropy sampling for all settings, since confidence threshold sampling incurs a larger performance drop on Dream than LLaDA. For the indicator score threshold used in Tab. 2, we adjust this hyperparameter around 0.9 for different datasets to ensure that our results are better than or comparable to confidence threshold sampling, thereby ensuring a fair comparison in terms of efficiency. For Dream model, we directly use 0.9 for the results in Tab. 4. In all trade-off curves, we select indicator score thresholds from 0.2 to 0.8 witn an interval 0.1 for the other data points.

# E    ANALYSIS OF INDICATOR PREDICTIONS

The speedup achieved by **NI Sampling** is in fact far below the speedup demonstrated in Sec. 3 (e.g., $4.8\times$ v.s. $11.4\times$ on GSM8K-256). To better understand the reason behind this, we investigate several practical cases on GSM8K dataset and identify several problems of the current neural indicator.

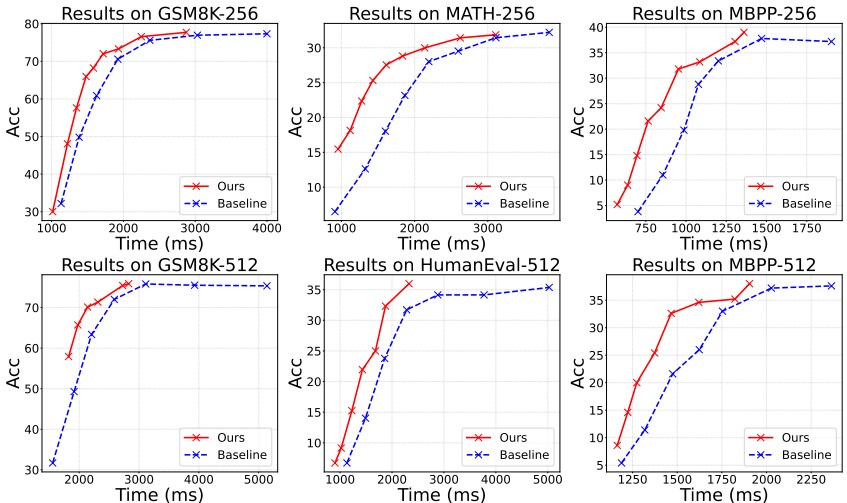

Figure 11: Performance-inference time trade-off curves of LLaDA-8B-Instruct.

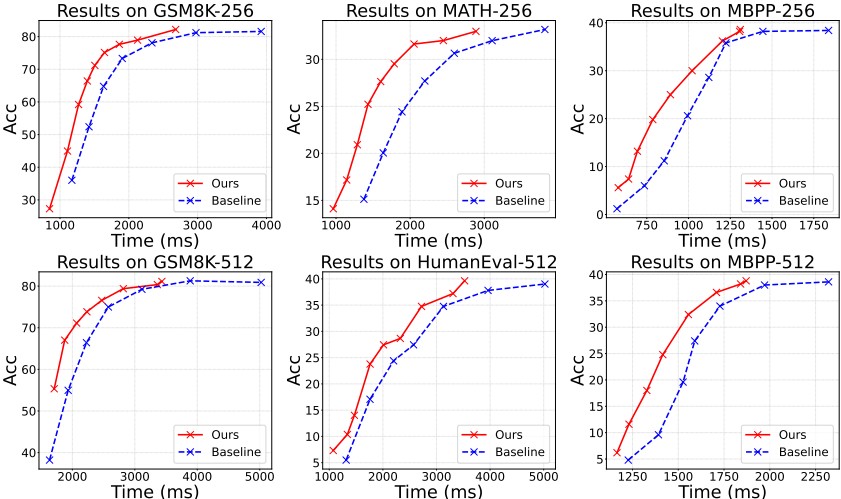

Figure 12: Performance-inference time trade-off curves of LLaDA-1.5.

**Our indicator tends to select tokens following an AR schedule,** even when ground truth trajectory-preserving-order does not follow such an order. There are two main impacts:

- The indicator can not identify correct tokens that are far away well. Examples are shown in Tab. 15. For GT trajectory preserving order, there are two zeros at later positions. However, NI sampling can only select the first zero.
- The indicator tends to select near tokens which actually can not be sampled with trajectory preserving order. Examples are shown in Tab. 16. We can see that trajectory preserving order doesn't sample the token "of" after "The cost", because it is not aligned with the final result "to" at this position. However, NI sampling selects this position to unmask, which changes the final result.

**Even though the ground truth trajectory-preserving-order follows an AR schedule, the indicator may just not select enough tokens.** Examples are shown in Tab. 17. NI sampling does not select correct tokens "he needs to" due to the low indicator score, although they follow AR schedule.

Although our ablation studies show that moderately increasing the parameter size does not significantly affect performance, we still hypothesize that these issues are because the model capacity or

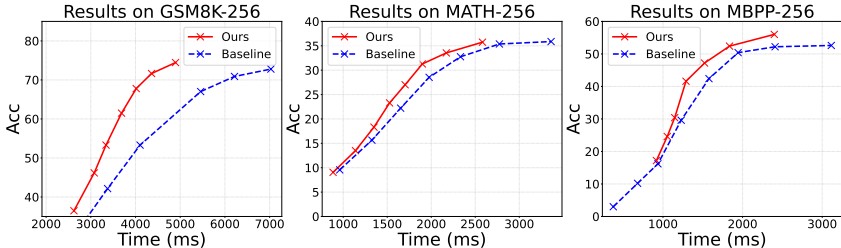

Figure 13: Performance-inference time trade-off curves of Dream-7B-Base.

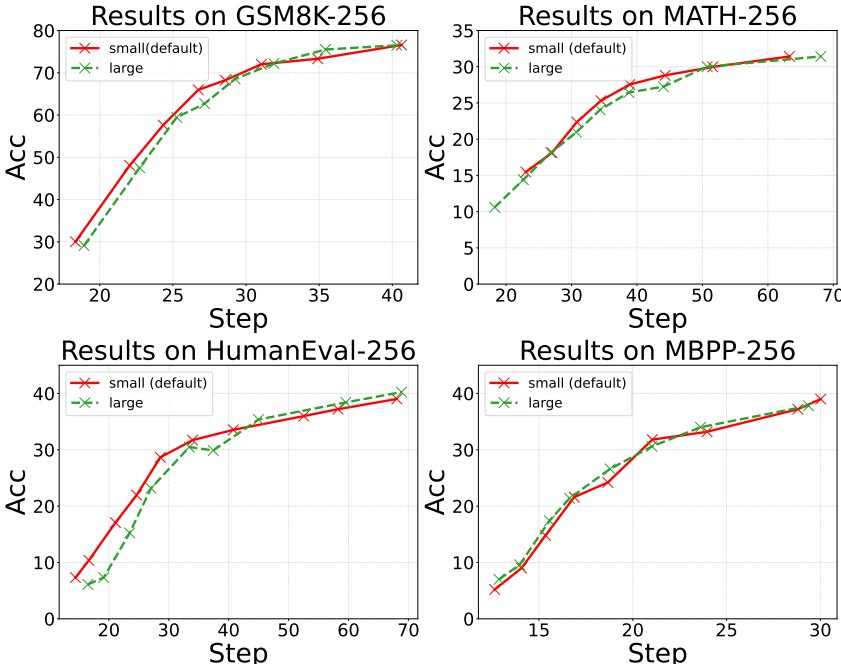

Figure 14: Ablation results on parameter size of the indicator.

the amount of training data is insufficient for the model to learn these difficult patterns. Since these patterns are too hard to learn, only scaling indicator size slightly is unlikely to be effective. We provide several potential solutions as follows:

- **Modifying the model architecture.** The current model is an MLP that makes independent predictions for each masked position, which is likely to be suboptimal and may prevent each position from sufficiently leveraging contextual information. A possible improvement is to replace the MLP with a Transformer-like architecture that makes joint decisions across all masked positions.

- **Scaling both model and data.** Scaling may be more effective after replacing model architecture.

- **Addressing the issue of accumulated errors.** In practical sampling, the errors introduced by the indicator accumulate over decoding steps, whereas the current training data does not account for this. Regenerating data using the indicator itself and retraining on such data may mitigate this issue.

- **Designing loss formulations and weighting strategies.** Modifying the loss formulation may allow the training to pay more attention on the non-semi-AR positions, which are more difficult for the model to learn.

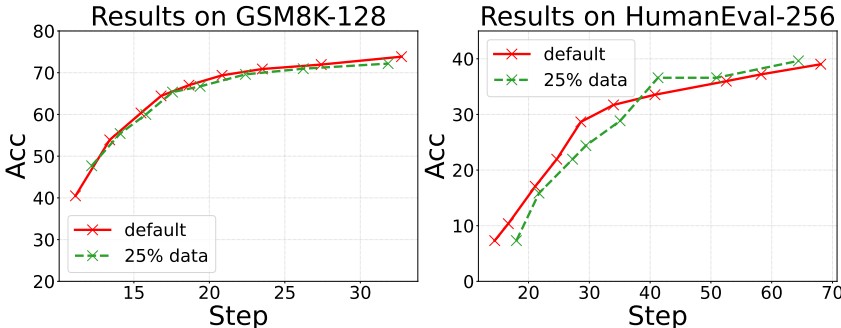

Figure 15: Ablation results on training set size.

Table 15: Examples of indicator prediction. "Original" means the sequence at the current step. "Trajectory Preserving Order" means conducting the current step with ground truth trajectory-preserving order. "Indicator" means coducting the current step with **NI Sampling**. [M] stands for the mask token. The differences between "Trajectory Preserving Order", "Indicator", and "Original" are marked as orange, while the key differences between "Trajectory Preserving Order" and "Indicator" are highlighted in red.

| Original | Trajectory Preserving Order | Indicator |
|---|---|---|
| Each implant has a base price of $2000.\n2. One implant has an additional cost of $500 for the crown, so the total cost for one implant is $2000 + $500 = $2500.\n3. George needs[M][M][M][M][M][M][M] [M][M][M][M][M][M][M][M][M]... (subsequent mask tokens are omitted) | 1. Each implant has a base price of $2000.\n2. One implant has an additional cost of $500 for the crown, so the total cost for one implant is $2000 + $500 = $2500.\n3. George needs 2 implants, so the total cost for the implants is[M]2[M][M][M][M][M][M][M][M][M][M]00[M][M][M][M][M]... | 1. Each implant has a base price of $2000.\n2. One implant has an additional cost of $500 for the crown, so the total cost for one implant is $2000 + $500 = $2500.\n3. George needs 2 implants, so the total cost for the implants is[M]2[M][M][M][M][M][M][M][M][M][M]0[M][M][M][M][M]... |

Table 16: Examples of indicator prediction. "Original" means the sequence at the current step. "Trajectory Preserving Order" means conducting the current step with ground truth trajectory-preserving order. "Indicator" means coducting the current step with **NI Sampling**. [M] stands for the mask token. The differences between "Trajectory Preserving Order", "Indicator", and "Original" are marked as orange, while the key differences between "Trajectory Preserving Order" and "Indicator" are highlighted in red

| Original | Trajectory Preserving Order | Indicator |
|---|---|---|
| The area of the bedroom is 18*12=≪18*12=216≫216 square feet \nThe cost of the new carpet is 216*12=$≪216*12=2592≫2592 \nThe cost of the[M] is[M]216[M][M]=$[M]2[M][M]* [M]=[M][M][M][M][M][M][M] [M][M][M][M][M][M][M]... | The area of the bedroom is 18*12=≪18*12=216≫216 square feet \nThe cost of the new carpet is 216*12=$≪216*12=2592≫2592 \nThe cost of the padding is $216*2=$≪216*2=432≫432\n The cost[M][M]the old[M][M] [M][M][M][M][M][M]=$[M][M] [M][M][M][M][M][M][M][M]... | The area of the bedroom is 18*12=≪18*12=216≫216 square feet \nThe cost of the new carpet is $216*12=$≪216*12=2592≫2592 \nThe cost of the padding is$216*2=$≪216*2=432≫432\n The cost of[M] the old[M][M][M] [M][M][M][M][M][M][M][M] [M][M][M][M]... |

## F DEMONSTRATION OF GENERATED EXAMPLES

In this section, we demonstrate examples with different sampling methods.

Table 17: Examples of indicator prediction. "Original" means the sequence at the current step. "Trajectory Preserving Order" means conducting the current step with ground truth trajectory-preserving order. "Indicator" means coducting the current step with **NI Sampling**. [M] stands for the mask token. The differences between "Trajectory Preserving Order", "Indicator", and "Original" are marked as orange, while the key differences between "Trajectory Preserving Order" and "Indicator" are highlighted in red

| Original | Trajectory Preserving Order | Indicator |
|---|---|---|
| Isaias plans to sell 3/5 of his 300 chickens, which is 3/5 * 300 = 180 chickens.\nTo make a profit of $2000, Isaias needs to sell the chickens at $50 per chicken, so he needs to earn 180 * $50 = $9000.\nTo make a profit of $2000, Isaias needs to earn $9000 - $2000 = $7000.\nSince Isaias needs to sell[M][M][M][M][M][M][M][M][M][M][M][M][M][M][M][M]... | Isaias plans to sell 3/5 of his 300 chickens, which is 3/5 * 300 = 180 chickens.\nTo make a profit of $2000, Isaias needs to sell the chickens at $50 per chicken, so he needs to earn 180 * $50 = $9000.\nTo make a profit of $2000, Isaias needs to earn $9000 - $2000 = $7000.\nSince Isaias needs to sellthe chickens at $50 per chicken, he needs to[M][M][M]000[M][M][M][M][M][M][M][M][M][M][M][M]... | Isaias plans to sell 3/5 of his 300 chickens, which is 3/5 * 300 = 180 chickens.\nTo make a profit of $2000, Isaias needs to sell the chickens at $50 per chicken, so he needs to earn 180 * $50 = $9000.\nTo make a profit of $2000, Isaias needs to earn $9000 - $2000 = $7000.\nSince Isaias needs to sell the chickens at $50 per chicken, [M][M][M][M][M][M][M][M][M][M][M][M][M][M][M][M]... |

Table 18: Qualitative examples across different sampling ethods. The **prompt** is: Janet's ducks lay 16 eggs per day. She eats three for breakfast every morning and bakes muffins for her friends every day with four. She sells the remainder at the farmers' market daily for $2 per fresh duck egg. How much in dollars does she make every day at the farmers' market?

| Full-step | Threshold | NI Sampling |
|---|---|---|
| Janet's ducks lay 16 eggs per day. She eats 3 eggs for breakfast every morning, so she has 16 - 3 = 13 eggs left. She bakes muffins for her friends with 4 eggs every day, so she has 13 - 4 = 9 eggs left. She sells the remaining eggs at the farmers' market for $2 per egg, so she makes 9 * $2 = $≪9*2=18≫18. Therefore, Janet makes $18 every day at the farmers' market. #### 18 (**NFE:128**) | Janet's ducks lay 16 eggs per day. She eats 3 eggs for breakfast every morning, so she has 16 - 3 = 13 eggs left. She bakes muffins for her friends with 4 eggs every day, so she has 13 - 4 = 9 eggs left. She sells the remaining eggs at the farmers' market for $2 per egg, so she makes 9 * $2 = $≪9*2=18≫18. Therefore, Janet makes $18 every day at the farmers' market. #### 18 (**NFE:49**) | Janet's ducks lay 16 eggs per day. She eats 3 eggs for breakfast every morning, so she has 16 - 3 = 13 eggs left. She bakes muffins for her friends with 4 eggs every day, so she has 13 - 4 = 9 eggs left. She sells the remaining eggs at the farmers' market for $2 per egg, so she makes 9 * $2 = $≪9*2=18≫18. Therefore, Janet makes $18 every day at the farmers' market. #### 18 (**NFE:34**) |

# G  THE USE OF LARGE LANGUAGE MODELS (LLMS)

We used LLM to support us in grammar refinement and language polishing. The paper was mainly written by the authors.

Table 19: Qualitative examples across different sampling ethods. The **prompt** is The great dragon, Perg, sat high atop mount Farbo, breathing fire upon anything within a distance of 1000 feet. Polly could throw the gold javelin, the only known weapon that could sleigh the dragon, for a distance of 400 feet, well within the reach of the dragon's flames. But when Polly held the sapphire gemstone, she could throw the javelin three times farther than when not holding the gemstone. If holding the gemstone, how far outside of the reach of the dragon's flames could Polly stand and still hit the dragon with the gold javelin?

| Full-step | Threshold | NI Sampling |
|---|---|---|
| When Polly was not holding the gemstone, she could throw the javelin 400 feet. When holding the gemstone, she could throw the javelin three times farther, so she could throw it 400 * 3 = 1200 feet. The dragon's flames could reach up to 1000 feet. Therefore, to still hit the dragon while holding the gemstone, Polly could stand 1200 - 1000 = 200 feet outside of the reach of the dragon's flames. #### 200 (**NFE:128**) | When Polly was not holding the gemstone, she could throw the javelin 400 feet. When holding the gemstone, she could throw the javelin three times farther, so she could throw it 400 * 3 = 1200 feet. The dragon's flames could reach a distance of 1000 feet. Therefore, when holding the gemstone, Polly could throw the javelin from 1200 - 1000 = 200 feet outside the reach of the dragon's flames. #### 200 (**NFE:54**) | When not holding the the the gemstone, she could throw the javelin 400 feet. When holding the gemstone, she could throw the javelin three times farther, so she could throw it 400 * 3 = 1200 feet. The dragon's flames could reach 1000 feet, so if she could throw the javelin 1200 feet, she could stand 1200 - 1000 = 200 feet outside of the reach of the dragon's flames. #### 200 (**NFE:38**) |

