# OpenReview forum: "NI Sampling: Accelerating Discrete Diffusion Sampling by Token Order Optimization"
_ICLR.cc/2026/Conference — ICLR 2026 Poster_

### Official Review · Reviewer_Duu3 · 2025-10-23

**Soundness:** 3
**Presentation:** 3
**Contribution:** 3
**Rating:** 6
**Confidence:** 3

**Summary:**

This paper studies acceleration methods for masked diffusion sampling. While masked diffusion models accept efficient prallel sampling/unmasking of multiple tokens without significant performance drop, how much speedup we can achieve has been unknown and perople used heuristic methods to determine the sampling order and parallelization magnitude. This paper introduces Trajectory-Preserving-Order, which measures to how much extent we can reduce the redundunt function evaluations of masked diffusion without changing the one-by-one sampling trajectory. In math and programming benchmarks with the top-1 probability argmax sampler, the authors confirm that up to 24x acceleration is theoretically possible. By learning the Trajectory-Preserving-Order by a neural network (neural indicator learning), NI sampling then achievs up to 14x acceleration, outperforming the threshold-based heuristic.

**Strengths:**

This paper is well written, and offers an evidence that the optimization of sampling positions in masked diffusion can greatly accelerate the sampling process, *even without changing the output/trajectory*, through the empirical studies with Trajectory-Preserving-Order. This offers a deeper understanding of masked diffusion sampling from a novel angle. The neural indicator learning also has a straightforward learning signal and succeeds in actually accelerating masked diffusions.

**Weaknesses:**

- [W1] A primary weakness is that this method works only for almost deterministic sampling strategies and tasks such as argmax sampling and math/programming tasks. Thus it does not align well with the direction of reproducing randomness, such as unconditional generation and creative tasks requiring diversity.
- [W2] As the authors admit in Secion 7, the performance of NI sampler in this paper is far from that of Trajectory-Prserving-Order and closer to the heuristic Threshold sampler without any training. I'm not sure how difficult learning the Trajectory-Preserving-Order is in reality.

Typos etc:
- L70: gsampling -> sampling
- L106: I find many `\cdots` for sequences (e.g., $1, \cdots, R$), but the standard notation aligning with amsmath (when the separator symbols are low) is `\ldots` (e.g., $1, \ldots, R$). It is not at all essential.
- L182: minize -> minimize
- L209: Preversing -> Preserving (same typo at **many** other positions as well)
- L272: Indicato -> Indicator

**Questions:**

While I lean to acceptance, I suspect that the NI sampler just conducts semi-AR sampling (that is not a major issue though). To better understand the method, I have the following question:
- [Q1] Trajectory-Preserving-Order may contain poisitions with a random coincidence such as comma and period. However, NI sampling in my intuition, when generalized well, follows mostly the natural order of words. That can result in a gap between the two, which might be very difficult to learn. I would like to see which word is sampled in which step, in examples like Tables 9 & 10, together with the result of Trajectory-Preserving-Order.

---

> ### Author Response · Authors · 2025-11-26
> **Response to Reviewer Duu3 1**
>
> - We sincerely thank your positive feedback and thoughtful questions! We are pleased to see your recognition of the novelty and potential of our trajectory-preserving-order, and the effectiveness of NI sampling. We respond to your concerns below. We are happy to discuss further if you still have questions.
>
> > ### W1: Only for almost deterministic sampling strategies and tasks such as argmax sampling and math/programming tasks
>
>   - First, we would like to emphasize that greedy sampling is generally acceptable in many practical scenarios, such as solving math problems. In these cases, our proposed approach offers substantial practical value. Moreover, the greedy sampling setting is widely adopted as the default configuration in recent works on discrete large language models—for example, LLaDA (see https://github.com/ML-GSAI/LLaDA/blob/570f29032d6824ea14977c89a8eb402e6eb25f96/generate.py#L154) and Fast-dllm (see https://github.com/NVlabs/Fast-dLLM), which introduces confidence-threshold sampling and serves as our primary baseline. Following this line of work, we choose greedy sampling as our primary evaluation setting.
>
>   - **Nevertheless, ensuring sampling diversity is crucial.** To address this concern, we provide additional experiments and discussion. Specifically, we first show that our current indicator does not degrade diversity. Then, by training a new indicator using random trajectories, we demonstrate that sampling diversity can be further improved. These results also confirm that our method does not overfit to deterministic trajectories.
>
>   - Firstly, our current indicator can be incorporated with random sampling easily without additional tuning.
>     - Method: specifically, similar to the implementation of confidence threshold sampling, we sample token at each position randomly according to the probability vector instead of using argmax. Then, we feed all actually sampled tokens together with its probability into the existing neural indicator and let it make decisions.
>     - Further discussion:
>       - **Our method follows the same paradigm as the random sampling implementation of confidence threshold sampling.** Specifically, under the greedy setting, both methods deterministically sample all tokens and keep the tokens whose scores exceed a threshold. When combined with random sampling, the modification remains exactly the same for both methods: deterministic selection is simply replaced by sampling from the predicted distribution. The only difference between the two methods lies in how eligible tokens are determined—confidence-threshold sampling uses confidence values, while our method uses the indicator score.
>       - **In fact, our method may be even more diverse than confidence threshold sampling.** At each step, the set of tokens that can be accepted under greedy sampling is larger in our method than in confidence-threshold sampling. Consequently, under random sampling, there is a higher probability of selecting tokens that deviate from the original trajectory. This increases the likelihood of trajectory divergence and thus leads to greater sampling diversity.
>
>   - The diversity of our method can be further enhanced by modifying the training process.
>     - Method: since our existing indicator has only seen tokens sampled via greedy sampling and deterministic trajectories, we revised the training strategy and retrained the model as follows:
>       - The deterministic trajectory generation process is replaced with a stochastic process using a temperature of 1.0. **This addresses the reviewer’s concern regarding potential overfitting to deterministic trajectories.**
>       - During training, tokens are sampled according to the model’s predicted probabilities rather than being selected solely with greedy strategy, and these sampled tokens serve as inputs to the indicator.

---

> ### Author Response · Authors · 2025-11-26
> **Response to Reviewer Duu3 2**
>
> (continued from previous response)
>
> - **Experiments.** We present experimental results of diversity evaluation to show that our existing neural indicator is fully compatible with stochastic sampling.
>     - **Setting and Evaluation.** We use LLaDA-8B-Instruct as the pre-trained model and evaluate on GSM8K and HumanEval. In addition, we include SQuAD Completion, SQuADv2, and TruthfulQA to cover a broader range of task types. We set the temperature to 1.0 for all experiments. For each problem, we generate $k$ different answers using multiple random seeds, and report the following metrics to quantify diversity:
>        - pass@k: A problem is considered correctly solved if at least one of the $k$ generated answers is correct. A diverse generative distribution should exhibit a steady increase in pass@k as $k$ grows. Note that we apply this metric to all datasets, not only code-related ones. For SQuADv2, whose official metric is the average F1 score, we compute the maximum F1 score among the $k$ independently generated answers for each problem and then report the average across all problems. We report pass@k under different values of $k$ and further compute the slope of the fitted line of pass@k with respect to $\log k$, which serves as our **primary metrics for evaluating diversity**. This metric is included following the suggestion of reviewer B3No.
>        - Self-BLEU [1]: For each answer, we compute its BLEU score with respect to the other $k-1$ answers. Then we take an average across all answers. We use both 1-gram-BLEU and 2-gram-BLEU. Lower values indicate higher diversity.
>        - Self-MAUVE [2]: We randomly split the $k$ answers into two groups and compute the MAUVE score between them. Lower values indicate higher diversity.
>
>         [1] Zhu et al., Texygen: A Benchmarking Platform for Text Generation Models.
>
>         [2] Pillutla et al., MAUVE: Measuring the Gap Between Neural Text and Human Text using Divergence Frontiers.
>
> - **Key Conclusions**
>   - For our existing indicator, experimental results are reported in the “old indicator” row of the following tables. The key conclusion is that **our existing indicator achieves comparable diversity to threshold sampling and full-step sampling across most datasets.** In most cases, the slope of pass@k with respect to $\log k$ is slightly higher than that of confidence threshold sampling, and similar to full-step sampling, indicating that our method achieves comparable diversity. Additionally, in most cases, our Self-BLEU and Self-MAUVE scores are comparable to or only slightly worse than those of confidence threshold sampling, further supporting the conclusion that the diversity of NI sampling is similar to baseline methods.
>
>   - For the indicator trained with random trajectories, experimental results are shown in the “new indicator” row of the following tables. **The key takeaway is that such indicator shows a larger slope of pass@k with respect to $\log k$, which indicates that training predictor on random trajectories can boost our sampling diversity.**
>
> - Overall, our method fully supports random sampling, and experimentally achieves performance comparable to full-step sampling, without overfitting to any specific deterministic trajectory. **This indicates that our approach is fully practical.**
>
> - We have updated the relevant content in Section C in the appendix of our revised paper.
>
> - Results are shown below.
>
>     - HumanEval-256 dataset
>
>     |               | Token/s | NFE   | Pass@1 | Pass@2 | Pass@4 | Pass@8 | Pass@16 | **Acc-log k slope$\uparrow$** |
>     |---------------|---------|-------|--------|--------|--------|--------|---------|---------------------------------|
>     | Full-step     | 42.91   | 256   | 36.99  | 44.69  | 51.43  | 57.81  | 63.76   | 6.67                            |
>     | Threshold     | 110.82  | 99.13 | 36.82  | 44.35  | 51.02  | 57.40  | 63.10   | 6.56                            |
>     | Old Indicator | 164.09  | 63.34 | 36.43  | 44.13  | 51.09  | 57.63  | 63.62   | 6.79                            |
>     | New Indicator | 155.78  | 66.72 | 36.70  | 44.38  | 51.34  | 58.03  | 64.33   | **6.89**                        |

---

> ### Author Response · Authors · 2025-11-26
> **Response to Reviewer Duu3 3**
>
> (continued from previous response)
>
> - Results are shown below.
>
>     - GSM8K-256 dataset
>
>     |               | Token/s | NFE   | Pass@1 | Pass@2 | Pass@4 | Pass@8 | **Acc-log k slope$\uparrow$** | Self-1-gram-BLEU$\downarrow$ | Self-2-gram-BLEU$\downarrow$ | Self-MAUVE$\downarrow$ |
>     |---------------|---------|-------|--------|--------|--------|--------|---------------------------------|--------------------------------|--------------------------------|--------------------------|
>     | Full-step     | 19.11   | 256   | 76.95  | 84.83  | 89.99  | 93.48  | 5.47                            | 0.916                          | 0.883                              | 0.966                    |
>     | Threshold     | 65.56   | 74.63 | 77.55  | 85.51  | 91.05  | 93.85  | 5.44                            | 0.920                          | 0.887                              | 0.974                    |
>     | Old Indicator | 86.74   | 54.70 | 77.78  | 85.97  | 91.05  | 93.70  | 5.28                            | 0.919                          | 0.886                              | 0.971                    |
>     | New Indicator | 82.36   | 57.68 | 77.33  | 84.91  | 90.83  | 94.47  | **5.73**                        | 0.918                          | 0.884                          | 0.973                    |
>
>     - Squad Completion
>
>     |               | Token/s | NFE   | pass@1 | pass@2 | pass@4 | pass@8 | **Acc-log k slope$\uparrow$** | self-1-gram-BLEU$\downarrow$ | self-2-gram-BLEU$\downarrow$ | Self-MAUVE$\downarrow$ |
>     |---------------|---------|-------|--------|--------|--------|--------|---------------------------------|--------------------------------|--------------------------------|--------------------------|
>     | full-step     | 48.48   | 128   | 78.3   | 83.3   | 88.2   | 91.3   | 4.39                            | 0.809                          | 0.600                          | 0.981                    |
>     | Threshold     | 174.64  | 34.03 | 77.5   | 84.0   | 87.0   | 89.5   | 3.90                            | 0.802                          | 0.600                          | 0.986                    |
>     | Old Indicator | 266.78  | 21.39 | 76.8   | 83.2   | 86.8   | 89.7   | 4.23                            | 0.810                          | 0.592                          | 0.982                    |
>     | New Indicator | 254.98  | 22.38 | 76.8   | 83.2   | 87.3   | 89.8   | **4.31**                        | 0.811                          | 0.589                          | 0.981                    |
>
>   - Squad V2
>
>    |               | Token/s | NFE   | pass@1 | pass@2 | pass@4 | pass@8 | **Acc-log k slope$\uparrow$** | Self-1-gram-BLEU$\downarrow$ | Self-2-gram-BLEU$\downarrow$ | Self-MAUVE$\downarrow$ |
>   |---------------|---------|-------|--------|--------|--------|--------|---------------------------------|--------------------------------|--------------------------------|--------------------------|
>     | full-step     | 47.94   | 128   | 27.64  | 31.76  | 37.46  | 41.92  | 4.85                            | 0.792                          | 0.710                          | 0.989                    |
>     | Threshold     | 202.27  | 31.23 | 28.05  | 33.21  | 37.92  | 42.21  | 4.72                            | 0.792                          | 0.710                          | 0.989                    |
>     | Old Indicator | 267.74  | 22.56 | 27.85  | 33.13  | 38.01  | 43.20  | 5.09                            | 0.797                          | 0.723                          | 0.989                    |
>     | New Indicator | 283.18  | 21.32 | 27.09  | 34.10  | 39.21  | 45.20  | **5.94**                        | 0.802                          | 0.720                          | 0.990                    |
>
>   - Truthful QA
>
>   |               | Token/s | NFE   | bleu_acc | rouge1_acc | rouge2_acc | rougeL_acc | Self-1-gram-BLEU$\downarrow$ | Self-2-gram-BLEU$\downarrow$ | Self-MAUVE$\downarrow$ |
>   |---------------|---------|-------|----------|------------|------------|------------|--------------------------------|--------------------------------|--------------------------|
>   | Full-step     | 46.54   | 128   | 53.86    | 53.61      | 49.07      | 52.88      | 0.898                          | 0.862                          | 0.983                    |
>   | Threshold     | 218.80  | 26.46 | 55.32    | 54.35      | 49.20      | 54.10      | 0.903                          | 0.867                          | 0.984                    |
>   | Old Indicator | 419.67  | 13.41 | 57.89    | 56.55      | 52.39      | 57.04      | 0.903                          | 0.867                          | 0.980                    |
>   | New Indicator | 408.94  | 13.90 | 61.57    | 59.00      | 54.59      | 59.73      | 0.901                          | 0.864                          | 0.985                    |

---

> ### Author Response · Authors · 2025-11-26
> **Response to Reviewer Duu3 4**
>
> > ### W2 & Q1: Not sure how difficult learning the Trajectory-Preserving-Order is & better to have further understanding
>
>   - Thank you for your thoughtful question! Following your advice, we investigate several practical cases on GSM8K dataset. We identify several representative issues below and **attach all examples at the end of the rebuttal, titled "Response to Reviewer Duu3 5: Examples of Indicator Prediction"**. We recommend that you see the examples in Section E of our revised paper, where we use different colors to highlight the difference.
>
>     1. While ground truth trajectory-preserving-order does not necessarily follow an autoregressive order, our indicator tends to select tokens following an autoregressive schedule, which has two main impacts:
>
>         - The indicator can not identify correct tokens that are far away well.
>
>         - The indicator tends to select near tokens which actually can not be sampled with trajectory preserving order.
>
>     2. Even though the ground truth trajectory-preserving-order follows an AR schedule, the indicator may just not select enough tokens.
>
>   - Although our ablation studies show that moderately increasing the parameter size does not significantly affect performance, we still hypothesize that these issues are because the model capacity or the amount of training data is insufficient for the model to learn these difficult patterns. Since these patterns are too hard to learn, only scaling indicator size slightly is unlikely to be effective. We list several potential solutions as follows:
>
>     - **Modifying the model architecture.** The current model is an MLP that makes independent predictions for each masked position, which is likely to be suboptimal and may prevent each position from sufficiently leveraging contextual information. A possible improvement is to replace the MLP with a Transformer-like architecture that makes joint decisions across all masked positions.
>
>     - **Scaling both model and data.** Scaling may be more effective after replacing model architecture.
>
>     - **Addressing the issue of accumulated errors.** In practical sampling, the errors introduced by the indicator accumulate over decoding steps, whereas the current training data does not account for this. Regenerating data using the indicator itself and retraining on such data may mitigate this issue.
>
>     - **Designing loss formulations and weighting strategies.** Modifying the loss formulation may allow the training to pay more attention on the non-semi-AR positions, which are more difficult for the model to learn.
>   - We have updated the relevant content in Section E in the appendix of our revised paper.
>
>   - Finally, we would like to clarify once again that revealing the upper bound of sampling-order optimization and proposing the NI Sampling method are two distinct contributions of our work. The upper bound represents a theoretical limit under idealized conditions.  In practice, due to constraints such as limited dataset size, neural network capacity, and training cost, this bound may not be attainable. Rather than aiming to exactly match this upper bound—which may be impossible under practical conditions—our goal is to develop a practical algorithm that effectively accelerates discrete diffusion models. Deriving a tighter upper bound that accounts for these practical constraints is an interesting direction for future work.
>
> > ### W3: Typos
>
>   - Thank you for pointing out these typos! We have fixed them in the revision of our paper.

---

> ### Author Response · Authors · 2025-11-26
> **Response to Reviewer Duu3 5: Examples of Indicator Prediction**
>
> - We attach examples of neural indicator prediction below, where "Original" denotes the input sequence at the current step, "GT Trajectory Preserving Order" denotes the decoding results of the current step with ground-truth trajectory-preserving-order, and "NI Sampling" denotes the decoding results with our method.
>
>     - Example 1: For GT trajectory preserving order, there are two zeros at later positions. However, NI sampling can only select the first zero. **We bold the corresponding tokens for trajectory-preserving-order and NI sampling.** Therefore, the conclusion is that **the indicator can not identify correct tokens that are far away well**.
>       - **Oringinal**: '1. Each implant has a base price of \\$2000.\\n2. One implant has an additional cost of \\$500 for the crown, so the total cost for one implant is \\$2000 + \\$500 = \\$2500.\\n3. George needs[M][M][M][M][M][M][M][M][M]...' (subsequent mask tokens are omitted)
>       - **GT Trajectory Preserving Order**: '1. Each implant has a base price of \\$2000.\\n2. One implant has an additional cost of \\$500 for the crown, so the total cost for one implant is \\$2000 + \\$500 = \\$2500.\\n3. George needs 2 implants, so the total cost for the implants is[M]2[M][M][M][M][M][M][M][M][M][M]$\textbf{00}$[M][M][M][M][M][M][M][M]...'
>       - **NI Sampling**: '1. Each implant has a base price of \\$2000.\\n2. One implant has an additional cost of \\$500 for the crown, so the total cost for one implant is \\$2000 + \\$500 = \\$2500.\\n3. George needs 2 implants, so the total cost for the implants is[M]2[M][M][M][M][M][M][M][M][M][M]$\textbf{0[M]}$[M][M][M][M][M][M][M][M]...'
>     - Example 2: We can see that trajectory preserving order doesn't sample the token "of" after "The cost", because it is not aligned with the final result "to" at this position. However, NI sampling selects this position to unmask, which changes the final result. **We bold the corresponding tokens for trajectory-preserving-order and NI sampling.** Therefore, the conclusion is that **the indicator tends to select near tokens which actually can not be sampled with trajectory preserving order**.
>       - **Original**: The area of the bedroom is 18\*12=<<18\*12=216>>216 square feet\\nThe cost of the new carpet is 216\*12=\\$<<216\*12=2592>>2592\\nThe cost of the[M] is[M]216[M][M]=\\$[M]2[M][M]\*[M]=[M][M][M][M][M][M][M][M][M][M][M][M][M][M][M][M][M][M][M]...
>       - **GT Trajectory Preserving Order**: The area of the bedroom is 18\*12=<<18\*12=216>>216 square feet\\nThe cost of the new carpet is 216\*12=\\$<<216\*12=2592>>2592\\nThe cost of the padding is \\$216\*2=\\$<<216\*2=432>>432\\nThe cost$\textbf{[M]}$[M] the old[M][M][M][M][M][M][M][M]=\\$[M][M][M][M][M][M][M][M]...
>       - **NI Sampling**: The area of the bedroom is 18\*12=<<18\*12=216>>216 square feet\\nThe cost of the new carpet is 216\*12=\\$<<216\*12=2592>>2592\\nThe cost of the padding is \\$216\*2=\\$<<216\*2=432>>432\\nThe cost $\textbf{of}$[M] the old[M][M][M][M][M][M][M][M][M][M][M][M][M][M][M][M]...
>     - Example 3: NI sampling does not select correct tokens "he needs to" due to the low indicator score, although they follow AR schedule. **We bold the corresponding tokens for trajectory-preserving-order and NI sampling.** Therefore, the conclusion is that **even though the ground truth trajectory-preserving-order follows an AR schedule, the indicator may just not select enough tokens**.
>       - **Original**: Isaias plans to sell 3/5 of his 300 chickens, which is 3/5 \* 300 = 180 chickens.\\nTo make a profit of \\$2000, Isaias needs to sell the chickens at \\$50 per chicken, so he needs to earn 180 \* \\$50 = \\$9000.\\nTo make a profit of \\$2000, Isaias needs to earn \\$9000 - \\$2000 = \\$7000.\\nSince Isaias needs to sell[M][M][M][M][M][M][M][M][M][M][M]...
>       - **GT Trajectory Preserving Order**: Isaias plans to sell 3/5 of his 300 chickens, which is 3/5 \* 300 = 180 chickens.\\nTo make a profit of \\$2000, Isaias needs to sell the chickens at \\$50 per chicken, so he needs to earn 180 \* \\$50 = \\$9000.\\nTo make a profit of \\$2000, Isaias needs to earn \\$9000 - \\$2000 = \\$7000.\\nSince Isaias needs to sell the chickens at \\$50 per chicken, $\textbf{he needs to}$[M][M][M]000[M][M][M][M][M][M][M][M][M]...
>       - **NI Sampling**: Isaias plans to sell 3/5 of his 300 chickens, which is 3/5 \* 300 = 180 chickens.\\nTo make a profit of \\$2000, Isaias needs to sell the chickens at \\$50 per chicken, so he needs to earn 180 \* \\$50 = \\$9000.\\nTo make a profit of \\$2000, Isaias needs to earn \\$9000 - \\$2000 = \\$7000.\\nSince Isaias needs to sell the chickens at \\$50 per chicken, $\textbf{[M][M][M]}$[M][M][M][M][M][M][M][M][M][M][M][M][M][M][M]...

---

> ### Comment · Reviewer_Duu3 · 2025-11-27
>
> Thank you for a thorough rebuttal. I think both answers to W1 and W2/Q1 address my concerns. Especially the steady improvement of pass@k slope convincingly shows the potential of NI sampler for random trajectories. Let me raise my score to 8.

---

> > ### Author Response · Authors · 2025-11-27
> > **Thank you for your feedback**
> >
> > We are glad to see our responses have addressed your concerns! Thank you once again for your insightful questions, which have been very helpful in improving the quality of our paper!

---

### Official Review · Reviewer_wmZ5 · 2025-10-27

**Soundness:** 1
**Presentation:** 4
**Contribution:** 2
**Rating:** 2
**Confidence:** 3

**Summary:**

This paper explores the potential for reducing sampling steps in discrete Large Language Models (dLLMs) by merging backward steps wherever possible. To approximate this sampling strategy, the authors propose Neural Indicator Sampling (NI Sampling), which learns to identify which positions can be unmasked at each step.

**Strengths:**

- The idea of merging multiple sampling steps into a single step is simple yet presents an interesting direction for accelerating generation.
- The proposed merging strategy significantly speeds up sampling while maintaining performance, at least in the scenario described in Section 3.

**Weaknesses:**

**Concerns about Diversity of Generated Samples**
- The approach relies heavily on a pre-computed, deterministic trajectory. I am concerned that this may lead to reduced diversity in the generated samples, especially if the sampling algorithm is entirely deterministic (see my question below).
- Whether generation distribution given by the merging algorithm is close to that given by the full sampling in a distribution sense is not discussed.
- While the experiments evaluate generation performance in terms of accuracy and speed, they do not assess diversity (e.g., entropy used in LLM field), which is a crucial aspect of generation quality.
- The generation tasks on mathematical and code datasets are insufficient for evaluating diversity.

**Performance Limitations of NI Sampling**
- It is not clear what are missing components to fill the significant performance gap between NI sampling and ideal sampling presented in Section 3. The ablation study on network size suggests that this gap is not due to the network's expressive power.
- Section 4.3 lists the network inputs, but it remains unclear what other components might be missing to fill the performance gap. It would be appreciated if insights into it can be more elaborated.

**Experiments**
- The proposed method is only compared with full sampling and the simple threshold-based method. However, there are related works [1,2] that should be discussed at least, even if they were originally proposed for vision domains.
- It would be beneficial to present Figure 4 and related figures as time vs. accuracy plots.

[1] Jose Lezama et al., "Improved Masked Image Generation with Token-Critic," ECCV 2022.
[2] Jose Lezama et al., "Discrete Predictor-Corrector Diffusion Models for Image Synthesis," ICLR 2023.

**Typos**
- "gsample" in Line 70

**Questions:**

- Is the proposed method limited to deterministic reference (trajectory)?
- Is the sampling procedure after which positions will be unmasked at each step is deterministic (based on top-1 sampling)?
- Could you clarify the statement "which may be due to the variance in evaluation" in Line 376. The meaning is unclear.

**Details Of Ethics Concerns:**

Potential ethical concerns are not discussed.

---

> ### Author Response · Authors · 2025-11-26
> **Response to Reviewer wmZ5 1**
>
> - Thank you for your constructive questions! We are pleased to see your recognition of merging adjacent steps as an interesting direction, and the effectiveness of NI sampling. We respond to your concerns below. We are happy to discuss further if you still have questions.
>
> > ### W1: Concerns about Diversity of Generated Samples
>
>   - First, we would like to emphasize that greedy sampling is generally acceptable in many practical scenarios, such as solving math problems. In these cases, our proposed approach offers substantial practical value. Moreover, the greedy sampling setting is widely adopted as the default configuration in recent works on discrete large language models—for example, LLaDA (see https://github.com/ML-GSAI/LLaDA/blob/570f29032d6824ea14977c89a8eb402e6eb25f96/generate.py#L154) and Fast-dllm (see https://github.com/NVlabs/Fast-dLLM), which introduces confidence-threshold sampling and serves as our primary baseline. Following this line of work, we choose greedy sampling as our primary evaluation setting.
>
>   - **Nevertheless, ensuring sampling diversity remains important.** Below, we provide additional discussions and experiments to address each of your concerns.
>
>   > #### W1.1 & Q1 & Q2: Does the method rely on top-1 sampling and deterministic trajectories? This reduces the sampling diversity
>
>   - Our method is actually not restricted to deterministic trajectories and in fact supports random sampling. Below, we first demonstrate that our current indicator naturally operates under stochastic sampling. We then show that, with alternative indicator training strategies, sampling diversity can be further improved.
>
> - Firstly, our current indicator can be incorporated with random sampling easily without additional tuning.
>     - Method: specifically, similar to the implementation of confidence threshold sampling, we sample token at each position randomly according to the probability vector instead of using argmax. Then, we feed all actually sampled tokens together with its probability into the existing neural indicator and let it make decisions.
>     - Further discussion:
>       - **Our method follows the same paradigm as the random sampling implementation of confidence threshold sampling.** Specifically, under the greedy setting, both methods deterministically sample all tokens and keep the tokens whose scores exceed a threshold. When combined with random sampling, the modification remains exactly the same for both methods: deterministic selection is simply replaced by sampling from the predicted distribution. The only difference between the two methods lies in how eligible tokens are determined—confidence-threshold sampling uses confidence values, while our method uses the indicator score.
>       - **In fact, our method may be even more diverse than confidence threshold sampling.** At each step, the set of tokens that can be accepted under greedy sampling is larger in our method than in confidence-threshold sampling. Consequently, under random sampling, there is a higher probability of selecting tokens that deviate from the original trajectory. This increases the likelihood of trajectory divergence and thus leads to greater sampling diversity.
>
> - The diversity of our method can be further enhanced by modifying the training process.
>   - Method: since our existing indicator has only seen tokens sampled via greedy sampling and deterministic trajectories, we revised the training strategy and retrained the model as follows:
>     - The deterministic trajectory generation process is replaced with a stochastic process using a temperature of 1.0. **This addresses the reviewer’s concern regarding potential overfitting to deterministic trajectories.**
>     - During training, tokens are sampled according to the model’s predicted probabilities rather than being selected solely with greedy strategy, and these sampled tokens serve as inputs to the indicator.

---

> ### Author Response · Authors · 2025-11-26
> **Response to Reviewer wmZ5 2**
>
> (continued from previous response)
>
> - **Experiments.** We present experimental results of diversity evaluation to show that our existing neural indicator is fully compatible with stochastic sampling.
>     - **Setting and Evaluation.** We use LLaDA-8B-Instruct as the pre-trained model and evaluate on GSM8K and HumanEval. In addition, we include SQuAD Completion, SQuADv2, and TruthfulQA to cover a broader range of task types. We set the temperature to 1.0 for all experiments. For each problem, we generate $k$ different answers using multiple random seeds, and report the following metrics to quantify diversity:
>        - pass@k: A problem is considered correctly solved if at least one of the $k$ generated answers is correct. A diverse generative distribution should exhibit a steady increase in pass@k as $k$ grows. Note that we apply this metric to all datasets, not only code-related ones. For SQuADv2, whose official metric is the average F1 score, we compute the maximum F1 score among the $k$ independently generated answers for each problem and then report the average across all problems. We report pass@k under different values of $k$ and further compute the slope of the fitted line of pass@k with respect to $\log k$, which serves as our **primary metrics for evaluating diversity**. This metric is included following the suggestion of reviewer B3No.
>        - Self-BLEU [1]: For each answer, we compute its BLEU score with respect to the other $k-1$ answers. Then we take an average across all answers. We use both 1-gram-BLEU and 2-gram-BLEU. Lower values indicate higher diversity.
>        - Self-MAUVE [2]: We randomly split the $k$ answers into two groups and compute the MAUVE score between them. Lower values indicate higher diversity.
>
>         [1] Zhu et al., Texygen: A Benchmarking Platform for Text Generation Models.
>
>         [2] Pillutla et al., MAUVE: Measuring the Gap Between Neural Text and Human Text using Divergence Frontiers.
>
> - **Key Conclusions**
>   - For our existing indicator, experimental results are reported in the “old indicator” row of the following tables. The key conclusion is that **our existing indicator achieves comparable diversity to threshold sampling and full-step sampling across most datasets.** In most cases, the slope of pass@k with respect to $\log k$ is slightly higher than that of confidence threshold sampling, and similar to full-step sampling, indicating that our method achieves comparable diversity. Additionally, in most cases, our Self-BLEU and Self-MAUVE scores are comparable to or only slightly worse than those of confidence threshold sampling, further supporting the conclusion that the diversity of NI sampling is similar to baseline methods.
>
>   - For the indicator trained with random trajectories, experimental results are shown in the “new indicator” row of the following tables. **The key takeaway is that such indicator shows a larger slope of pass@k with respect to $\log k$, which indicates that training predictor on random trajectories can boost our sampling diversity.**
>
> - Overall, our method fully supports random sampling, and experimentally achieves performance comparable to full-step sampling, without overfitting to any specific deterministic trajectory. **This indicates that our approach is fully practical.**
>
> - We have updated the relevant content in Section C in the appendix of our revised paper.
>
> - Results are shown below.
>
>     - HumanEval-256 dataset
>
>     |               | Token/s | NFE   | Pass@1 | Pass@2 | Pass@4 | Pass@8 | Pass@16 | **Acc-log k slope$\uparrow$** |
>     |---------------|---------|-------|--------|--------|--------|--------|---------|---------------------------------|
>     | Full-step     | 42.91   | 256   | 36.99  | 44.69  | 51.43  | 57.81  | 63.76   | 6.67                            |
>     | Threshold     | 110.82  | 99.13 | 36.82  | 44.35  | 51.02  | 57.40  | 63.10   | 6.56                            |
>     | Old Indicator | 164.09  | 63.34 | 36.43  | 44.13  | 51.09  | 57.63  | 63.62   | 6.79                            |
>     | New Indicator | 155.78  | 66.72 | 36.70  | 44.38  | 51.34  | 58.03  | 64.33   | **6.89**                        |

---

> ### Author Response · Authors · 2025-11-26
> **Response to Reviewer wmZ5 3**
>
> (continued from previous response)
>
> - Results are shown below.
>
>     - GSM8K-256 dataset
>
>     |               | Token/s | NFE   | Pass@1 | Pass@2 | Pass@4 | Pass@8 | **Acc-log k slope$\uparrow$** | Self-1-gram-BLEU$\downarrow$ | Self-2-gram-BLEU$\downarrow$ | Self-MAUVE$\downarrow$ |
>     |---------------|---------|-------|--------|--------|--------|--------|---------------------------------|--------------------------------|--------------------------------|--------------------------|
>     | Full-step     | 19.11   | 256   | 76.95  | 84.83  | 89.99  | 93.48  | 5.47                            | 0.916                          | 0.883                              | 0.966                    |
>     | Threshold     | 65.56   | 74.63 | 77.55  | 85.51  | 91.05  | 93.85  | 5.44                            | 0.920                          | 0.887                              | 0.974                    |
>     | Old Indicator | 86.74   | 54.70 | 77.78  | 85.97  | 91.05  | 93.70  | 5.28                            | 0.919                          | 0.886                              | 0.971                    |
>     | New Indicator | 82.36   | 57.68 | 77.33  | 84.91  | 90.83  | 94.47  | **5.73**                        | 0.918                          | 0.884                          | 0.973                    |
>
>     - Squad Completion
>
>     |               | Token/s | NFE   | pass@1 | pass@2 | pass@4 | pass@8 | **Acc-log k slope$\uparrow$** | self-1-gram-BLEU$\downarrow$ | self-2-gram-BLEU$\downarrow$ | Self-MAUVE$\downarrow$ |
>     |---------------|---------|-------|--------|--------|--------|--------|---------------------------------|--------------------------------|--------------------------------|--------------------------|
>     | full-step     | 48.48   | 128   | 78.3   | 83.3   | 88.2   | 91.3   | 4.39                            | 0.809                          | 0.600                          | 0.981                    |
>     | Threshold     | 174.64  | 34.03 | 77.5   | 84.0   | 87.0   | 89.5   | 3.90                            | 0.802                          | 0.600                          | 0.986                    |
>     | Old Indicator | 266.78  | 21.39 | 76.8   | 83.2   | 86.8   | 89.7   | 4.23                            | 0.810                          | 0.592                          | 0.982                    |
>     | New Indicator | 254.98  | 22.38 | 76.8   | 83.2   | 87.3   | 89.8   | **4.31**                        | 0.811                          | 0.589                          | 0.981                    |
>
>   - Squad V2
>
>    |               | Token/s | NFE   | pass@1 | pass@2 | pass@4 | pass@8 | **Acc-log k slope$\uparrow$** | Self-1-gram-BLEU$\downarrow$ | Self-2-gram-BLEU$\downarrow$ | Self-MAUVE$\downarrow$ |
>   |---------------|---------|-------|--------|--------|--------|--------|---------------------------------|--------------------------------|--------------------------------|--------------------------|
>     | full-step     | 47.94   | 128   | 27.64  | 31.76  | 37.46  | 41.92  | 4.85                            | 0.792                          | 0.710                          | 0.989                    |
>     | Threshold     | 202.27  | 31.23 | 28.05  | 33.21  | 37.92  | 42.21  | 4.72                            | 0.792                          | 0.710                          | 0.989                    |
>     | Old Indicator | 267.74  | 22.56 | 27.85  | 33.13  | 38.01  | 43.20  | 5.09                            | 0.797                          | 0.723                          | 0.989                    |
>     | New Indicator | 283.18  | 21.32 | 27.09  | 34.10  | 39.21  | 45.20  | **5.94**                        | 0.802                          | 0.720                          | 0.990                    |
>
>   - Truthful QA
>
>   |               | Token/s | NFE   | bleu_acc | rouge1_acc | rouge2_acc | rougeL_acc | Self-1-gram-BLEU$\downarrow$ | Self-2-gram-BLEU$\downarrow$ | Self-MAUVE$\downarrow$ |
>   |---------------|---------|-------|----------|------------|------------|------------|--------------------------------|--------------------------------|--------------------------|
>   | Full-step     | 46.54   | 128   | 53.86    | 53.61      | 49.07      | 52.88      | 0.898                          | 0.862                          | 0.983                    |
>   | Threshold     | 218.80  | 26.46 | 55.32    | 54.35      | 49.20      | 54.10      | 0.903                          | 0.867                          | 0.984                    |
>   | Old Indicator | 419.67  | 13.41 | 57.89    | 56.55      | 52.39      | 57.04      | 0.903                          | 0.867                          | 0.980                    |
>   | New Indicator | 408.94  | 13.90 | 61.57    | 59.00      | 54.59      | 59.73      | 0.901                          | 0.864                          | 0.985                    |

---

> ### Author Response · Authors · 2025-11-26
> **Response to Reviewer wmZ5 4**
>
> > #### W1.3 & W1.4: Lack of diversity evaluation and other datasets
>
>    - As discussed above, we have provided diversity evaluation results using pass@k, Self-BLEU, and Self-MAUVE, along with additional datasets such as TruthfulQA, SQuAD Completion, and SQuADv2. If further experiments are desired, we are happy to include them.
>
> > #### W1.2: Whether our method is close to full-step sampling in a distribution sense
>
>    - **For the greedy sampling case**, as discussed in Sec. 3.2, merging adjacent steps along the trajectory does not affect the final sampling result. Specifically, for the original sampling trajectory, we merge the subsequent step into the current step only when the current model prediction at all positions in the subsequent step align with the final results. In this way, sampling these two steps simultaneously incurs no impact on later steps. Therefore, merging steps will never harm the sampling results of the original trajectory.
>
>    - **For random sampling**, firstly, we would like to emphasize the empirical evidence across multiple datasets and multiple metrics demonstrating that our method achieves diversity and accuracy similar to baseline methods, and is thus of strong practical utility.
>
>   - To fully address the reviewer's concern, we further provide theoretical analysis regarding the distribution loss of merging two adjacent steps along the trajectory in the next part of our rebuttal.

---

> ### Author Response · Authors · 2025-11-26
> **Response to Reviewer wmZ5 5**
>
> (continued from previous response)
>
> - In this analysis, we mainly follow the discussion in the paper of confidence threshold sampling [1]. We assume the probability given by the pre-trained discrete diffusion model is accurate.
>
>     - **Problem definition and notations.** Since our training process uses confidence threshold sampling as the base sampling method to generate our reference trajectory, we analyze step merging under this setting. Under the greedy setting, assuming the threshold of the base sampling method is $1-\epsilon$, the reference trajectory decodes $n_1$ tokens at the current step, and will decode $n_2$ tokens at the next step. We denote the random variable of the first group as $X_1$ and greedy sampling result $x_1$, and the random variable of the second group as $X_2$ and greedy sampling result $x_2$. Now our NI sampling will decode $X_1$ and $X_2$ simultaneously at the current step. We denote the minimum confidence of $X_2$ at the current step as $1-\epsilon^{\prime}$. We denote $p(X_1, X_2)$ as the distribution of sampling $X_1$ and $X_2$ in two separate steps, respectively; $q(X_1, X_2)$ as the distribution of sampling $X_1$ and $X_2$ in a single merged step. We calculate $L_p$ distance between these two distributions as the metric to evaluate the distribution gap.
>
>     - We first give the bound of $|p((X_1,X_2)=(x_1,x_2))-q((X_1,X_2)=(x_1,x_2))|$
>
>       - $p((X_1,X_2)=(x_1,x_2))=p(X_1=x_1)p(X_2=x_2|X_1=x_1)$. Consider $p(X_1=x_1)$, which can be given as $P(X_1=x_1)=\prod_{j=1}^{n_1}(1-\epsilon_j)$, where $1-\epsilon_j$ is the confidence of the j-th token. The bound of $P(X_1=x_1)$ can be given as:
>
>            $1-n_1\epsilon\leq1-\sum_{j=1}^{n_1}\epsilon_j\leq P(X_1=x_1)\leq 1-\epsilon$
>
>       - Similarly, the bound of $P(X_2=x_2|X_1=x_1)$ can be given as:
>
>            $1-n_2\epsilon\leq1-\sum_{j=1}^{n_2}\epsilon_j\leq P(X_2=x_2|X_1=x_1)\leq 1-\epsilon$
>
>       - So the bound of $p((X_1,X_2)=(x_1,x_2))$ is:
>
>            $(1-n_1\epsilon)(1-n_2\epsilon)\leq p((X_1,X_2)=(x_1,x_2))\leq (1-\epsilon)^2$
>
>     - Then we calculate the bound of $q((X_1,X_2)=(x_1,x_2))$. In a similar way, the bound can be given as:
>
>          $1-n_1\epsilon-n_2\epsilon^{\prime}\leq q((X_1,X_2)=(x_1,x_2))\leq 1-\epsilon^{\prime}$
>
>     - So $|p((X_1,X_2)=(x_1,x_2))-q((X_1,X_2)=(x_1,x_2))|\leq max(|1-\epsilon^{\prime}-(1-n_1\epsilon)(1-n_2\epsilon)|,|(1-\epsilon)^2+n_1\epsilon+n_2\epsilon^{\prime}-1|)$
>
>     - For $\sum_{(z_1,z_2)\neq(x_1,x_2)}|p(z_1,z_2)-q(z_1,z_2)|$, its bound can be given as:
>
>          $\sum_{(z_1,z_2)\neq(x_1,x_2)}|p(z_1,z_2)-q(z_1,z_2)|\leq p((X_1,X_2)\neq(x_1,x_2))+q((X_1,X_2)\neq(x_1,x_2))\leq 2n_1\epsilon+n_2(\epsilon^{\prime}+\epsilon)-n_1n_2\epsilon^2$
>
>     - We can easily get that for all $(z_1,z_2)\neq(x_1,x_2)$, $|p(z_1,z_2)-q(z_1,z_2)|\leq\epsilon$
>
>     - Thus, $\sum_{(z_1,z_2)\neq(x_1,x_2)}|p(z_1,z_2)-q(z_1,z_2)|^p\leq(max_{(z_1,z_2)\neq(x_1,x_2)}|p(z_1,z_2)-q(z_1,z_2)|)^{p-1}\sum_{(z_1,z_2)\neq(x_1,x_2)}|p(z_1,z_2)-q(z_1,z_2)|\leq\epsilon^{p-1}(2n_1\epsilon+n_2(\epsilon^{\prime}+\epsilon)-n_1n_2\epsilon^2)$
>
>     - Now we can give the bound of $L_p$ distance: $$L_p(p,q)=(|p((X_1,X_2)=(x_1,x_2))-q((X_1,X_2)=(x_1,x_2))|^p+\sum_{(z_1,z_2)\neq(x_1,x_2)}|p(z_1,z_2)-q(z_1,z_2)|^p)^{1/p}\leq (max(|1-\epsilon^{\prime}-(1-n_1\epsilon)(1-n_2\epsilon)|^p,|(1-\epsilon)^2+n_1\epsilon+n_2\epsilon^{\prime}-1|^p)+\epsilon^{p-1}(2n_1\epsilon+n_2(\epsilon^{\prime}+\epsilon)-n_1n_2\epsilon^2))^{1/p}$$
>
>     - Since all tokens in $X_2$ should not be sampled in the first step, so $\epsilon^{\prime}>\epsilon$; Besides, since $X_2$ should be sampled in the second step, so $1-\epsilon^{\prime}\geq(1-\epsilon)(1-n_1\epsilon)$.  We can further substitute this into the above equation to eliminate $\epsilon^{\prime}$: $$L_p(p,q)\leq (max(|1-\epsilon-(1-n_1\epsilon)(1-n_2\epsilon)|^p,|-\epsilon^2(n_2n_1-1)+(n_1n_2+n_1+n_2-2)\epsilon|^p)+\epsilon^{p-1}((2(n_1+n_2)+n_1n_2)\epsilon-2n_1n_2\epsilon^2))^{1/p}=(|-\epsilon^2(n_2n_1-1)+(n_1n_2+n_1+n_2-2)\epsilon|^p+\epsilon^{p-1}((2(n_1+n_2)+n_1n_2)\epsilon-2n_1n_2\epsilon^2))^{1/p}$$
>
> [1] Wu et al., Fast-dLLM: Training-free Acceleration of Diffusion LLM by Enabling KV Cache and Parallel Decoding

---

> ### Author Response · Authors · 2025-11-26
> **Response to Reviewer wmZ5 6**
>
> > ### W2: Performance Limitations of NI Sampling
>
>   - We would like to first clarify that **revealing the upper bound of token-order optimization** and **proposing NI-Sampling to effectively reduce the number of steps** are two distinct contributions of our work. The former not only motivates the latter but also provides insights that may inspire future research, while the latter serves as a practically meaningful acceleration method. The upper bound represents a theoretical limit under idealized conditions.  In practice, due to constraints such as limited dataset size, neural network capacity, and training cost, this bound may not be attainable. Rather than aiming to exactly match this upper bound—which may be impossible under practical conditions—our goal is to develop a practical algorithm that effectively accelerates discrete diffusion models. Deriving a tighter upper bound that accounts for these practical constraints is an interesting direction for future work.
>
>   - Nevertheless, to better understand the limitations of the indicator, we examine several practical cases on the GSM8K dataset and identify some issues with the current neural indicator. We summarize the issues below and **attach all examples at the end of the rebuttal, titled "Response to Reviewer wmZ5 8: Examples of Indicator Prediction"**. We recommend that you see the examples in Section E of our revised paper, where we use different colors to highlight the difference.
>
>     - While ground truth trajectory-preserving-order does not necessarily follow an autoregressive order, our indicator tends to select tokens following an autoregressive schedule, which has two main impacts:
>
>       - The indicator can not identify correct tokens that are far away well.
>
>       - The indicator tends to select near tokens which actually can not be sampled with trajectory preserving order.
>
>     - Even though the ground truth trajectory-preserving-order follows an AR schedule, the indicator may just not select enough tokens.
>
>   - Although our ablation studies show that moderately increasing the parameter size does not significantly affect performance, we still hypothesize that these issues are because the model capacity or the amount of training data is insufficient for the model to learn these difficult patterns. Since these patterns are too hard to learn, only scaling indicator size slightly is unlikely to be effective. We list several potential solutions as follows:
>
>     - **Modifying the model architecture.** The current model is an MLP that makes independent predictions for each masked position, which is likely to be suboptimal and may prevent each position from sufficiently leveraging contextual information. A possible improvement is to replace the MLP with a Transformer-like architecture that makes joint decisions across all masked positions.
>
>     - **Scaling both model and data.** Scaling may be more effective after replacing model architecture.
>
>     - **Addressing the issue of accumulated errors.** In practical sampling, the errors introduced by the indicator accumulate over decoding steps, whereas the current training data does not account for this. Regenerating data using the indicator itself and retraining on such data may mitigate this issue.
>
>     - **Designing loss formulations and weighting strategies.** Modifying the loss formulation may allow the training to pay more attention on the non-semi-AR positions, which are more difficult for the model to learn.
>   - We have updated the relevant content in Section E in the appendix of our revised paper.

---

> ### Author Response · Authors · 2025-11-26
> **Response to Reviewer wmZ5 7**
>
> > ### W3: Experiments
> > #### W3.1: Discussion on more related works
>   - Thank you for pointing out these two methods for us! They all use a neural network to decide which token to remask, which is conceptually close to our method. The difference is that they target at better performance of visual discrete diffusion models, while our method focus on accelerating dLLMs. We will discuss these works in the related work as below:
>     - Token-Critic and DPC also apply an extra neural network to decide which token to remask at each step. However, they aim for better performance of image MDMs, which is different from our work.
>     - We update the relevant content in Section 6.2 of our revised paper.
>   - To further address your concern, we conduct experiments to compare these methods. Since DPC [2] uses more corrector steps than Token-Critic[1], which is worse for acceleration, we only compare our method with Token-Critic [1].
>     - Setting: for fair comparison, we use the same neural indicator and training setting for our method and Token-Critic. Following its paper, we use top-k sampling for Token-Critic and combine it with confidence threshold sampling as NI sampling does. Results on GSM8K are shown below. We can see that Token-Critic can not outperform NI sampling in the acceleration case.
>
>     ||||||
>     |-|-|-|-|-|
>     |NI Sampling|NFE|15.47|20.84|27.41|
>     |NI Sampling|Acc|60.35|69.37|71.95|
>     |Token-Critic|NFE|15.08 (Top-8)|22.01 (Top-4)| 28.58 (Top-2)|
>     |Token-Critic|Acc|33.59|42.84|53.83|
>
>      [1] Jose Lezama et al., "Improved Masked Image Generation with Token-Critic," ECCV 2022.
>
>      [2] Jose Lezama et al., "Discrete Predictor-Corrector Diffusion Models for Image Synthesis," ICLR 2023.
>
> > #### W3.2: Present Figure 4 and related figures as time vs. accuracy plots.
>   - Thank you for your suggestion! We have revised our paper according to your suggestion.
>
> > ### W4: Typos
>
>   - Thank you for pointing out this typo! We have fixed it in the revision of our paper.
>
> >### Q3：clarify the statement "which may be due to the variance in evaluation" in Line 376
>   - Since the indicator inevitably incurs some loss in training, its sampling results cannot be identical to the base trajectories. By this statement, we mean the impact of such difference on sampling is random. At this stage, the predictor is already able to largely preserve performance, so when the differences are favorable, the indicator may even outperform full-step or threshold sampling. Moreover, we observed that confidence threshold sampling occasionally also performs better than full-step sampling, as can be found in Table 2 of our paper and [3], which may be attributed to a similar effect.
>
>     [3] Wu et al., Fast-dLLM: Training-free Acceleration of Diffusion LLM by Enabling KV Cache and Parallel Decoding

---

> ### Author Response · Authors · 2025-11-26
> **Response to Reviewer wmZ5 8: Examples of Indicator Prediction**
>
> - We attach examples of neural indicator prediction below, where "Original" denotes the input sequence at the current step, "GT Trajectory Preserving Order" denotes the decoding results of the current step with ground-truth trajectory-preserving-order, and "NI Sampling" denotes the decoding results with our method.
>
>     - Example 1: For GT trajectory preserving order, there are two zeros at later positions. However, NI sampling can only select the first zero. **We bold the corresponding tokens for trajectory-preserving-order and NI sampling.** Therefore, the conclusion is that **the indicator can not identify correct tokens that are far away well**.
>       - **Oringinal**: '1. Each implant has a base price of \\$2000.\\n2. One implant has an additional cost of \\$500 for the crown, so the total cost for one implant is \\$2000 + \\$500 = \\$2500.\\n3. George needs[M][M][M][M][M][M][M][M][M]...' (subsequent mask tokens are omitted)
>       - **GT Trajectory Preserving Order**: '1. Each implant has a base price of \\$2000.\\n2. One implant has an additional cost of \\$500 for the crown, so the total cost for one implant is \\$2000 + \\$500 = \\$2500.\\n3. George needs 2 implants, so the total cost for the implants is[M]2[M][M][M][M][M][M][M][M][M][M]$\textbf{00}$[M][M][M][M][M][M][M][M]...'
>       - **NI Sampling**: '1. Each implant has a base price of \\$2000.\\n2. One implant has an additional cost of \\$500 for the crown, so the total cost for one implant is \\$2000 + \\$500 = \\$2500.\\n3. George needs 2 implants, so the total cost for the implants is[M]2[M][M][M][M][M][M][M][M][M][M]$\textbf{0[M]}$[M][M][M][M][M][M][M][M]...'
>     - Example 2: We can see that trajectory preserving order doesn't sample the token "of" after "The cost", because it is not aligned with the final result "to" at this position. However, NI sampling selects this position to unmask, which changes the final result. **We bold the corresponding tokens for trajectory-preserving-order and NI sampling.** Therefore, the conclusion is that **the indicator tends to select near tokens which actually can not be sampled with trajectory preserving order**.
>       - **Original**: The area of the bedroom is 18\*12=<<18\*12=216>>216 square feet\\nThe cost of the new carpet is 216\*12=\\$<<216\*12=2592>>2592\\nThe cost of the[M] is[M]216[M][M]=\\$[M]2[M][M]\*[M]=[M][M][M][M][M][M][M][M][M][M][M][M][M][M][M][M][M][M][M]...
>       - **GT Trajectory Preserving Order**: The area of the bedroom is 18\*12=<<18\*12=216>>216 square feet\\nThe cost of the new carpet is 216\*12=\\$<<216\*12=2592>>2592\\nThe cost of the padding is \\$216\*2=\\$<<216\*2=432>>432\\nThe cost$\textbf{[M]}$[M] the old[M][M][M][M][M][M][M][M]=\\$[M][M][M][M][M][M][M][M]...
>       - **NI Sampling**: The area of the bedroom is 18\*12=<<18\*12=216>>216 square feet\\nThe cost of the new carpet is 216\*12=\\$<<216\*12=2592>>2592\\nThe cost of the padding is \\$216\*2=\\$<<216\*2=432>>432\\nThe cost $\textbf{of}$[M] the old[M][M][M][M][M][M][M][M][M][M][M][M][M][M][M][M]...
>     - Example 3: NI sampling does not select correct tokens "he needs to" due to the low indicator score, although they follow AR schedule. **We bold the corresponding tokens for trajectory-preserving-order and NI sampling.** Therefore, the conclusion is that **even though the ground truth trajectory-preserving-order follows an AR schedule, the indicator may just not select enough tokens**.
>       - **Original**: Isaias plans to sell 3/5 of his 300 chickens, which is 3/5 \* 300 = 180 chickens.\\nTo make a profit of \\$2000, Isaias needs to sell the chickens at \\$50 per chicken, so he needs to earn 180 \* \\$50 = \\$9000.\\nTo make a profit of \\$2000, Isaias needs to earn \\$9000 - \\$2000 = \\$7000.\\nSince Isaias needs to sell[M][M][M][M][M][M][M][M][M][M][M]...
>       - **GT Trajectory Preserving Order**: Isaias plans to sell 3/5 of his 300 chickens, which is 3/5 \* 300 = 180 chickens.\\nTo make a profit of \\$2000, Isaias needs to sell the chickens at \\$50 per chicken, so he needs to earn 180 \* \\$50 = \\$9000.\\nTo make a profit of \\$2000, Isaias needs to earn \\$9000 - \\$2000 = \\$7000.\\nSince Isaias needs to sell the chickens at \\$50 per chicken, $\textbf{he needs to}$[M][M][M]000[M][M][M][M][M][M][M][M][M]...
>       - **NI Sampling**: Isaias plans to sell 3/5 of his 300 chickens, which is 3/5 \* 300 = 180 chickens.\\nTo make a profit of \\$2000, Isaias needs to sell the chickens at \\$50 per chicken, so he needs to earn 180 \* \\$50 = \\$9000.\\nTo make a profit of \\$2000, Isaias needs to earn \\$9000 - \\$2000 = \\$7000.\\nSince Isaias needs to sell the chickens at \\$50 per chicken, $\textbf{[M][M][M]}$[M][M][M][M][M][M][M][M][M][M][M][M][M][M][M]...

---

> > ### Comment · Reviewer_wmZ5 · 2025-11-27
> >
> > Thank you for your detailed rebuttal. I have carefully reviewed all aspects of your response, including the updates made to the manuscript. Your replies address my mojar concernes, particularly those regarding generation performance in terms of diversity, the discussion of the proposed algorithm with respect to the achieved distribution, and the addition of a baseline. Although the underlying causes of the significant performance gap compared to the ideal sampling case remain somewhat unclear, your list of potential reasons is valuable and will likely inform future research. Accordingly, I have decided to increase my score to 6.
> >
> > I have one question and one suggestion regarding your response and the revised manuscript:
> >
> > - You mentioned that you incorporated my comments on Figure 4, but I was unable to locate the corresponding update in the revised manuscript. Could you please clarify which figure reflects this modification?
> > - In my view, reporting a comparison between the proposed method and Token-Critic remains informative, even though Token-Critic was originally proposed for image generation. Would it be possible to include these results somewhere in the manuscript, for example in the appendix?

---

> > > ### Author Response · Authors · 2025-11-27
> > > **Response to Reviewer wmZ5**
> > >
> > > Thank you for your time and positive feedback! We are glad to hear that our responses have addressed most of your concerns. Regarding your further questions and suggestions:
> > >
> > > -  We sincerely apologize for having uploaded an incorrect version of our paper. Following your suggestion, we have updated the horizontal axes of Fig. 4, 7, 8, 9 to actual running time based on your suggestion. The figures look similar to the previous version because the computational overhead of the indicator is negligible. Please refer to the latest revision.
> > >
> > > -  Thank you for your valuable suggestion. Due to the limited space in the main text, we have added the corresponding discussion and results in Section D.1 in the appendix of our revised paper.
> > >
> > > We truly appreciate your insightful comments, which have helped us further improve the quality of the paper. Please feel free to let us know if any additional clarification is needed. Happy to discuss more!

---

### Official Review · Reviewer_B3No · 2025-11-01

**Soundness:** 1
**Presentation:** 3
**Contribution:** 2
**Rating:** 2
**Confidence:** 3

**Summary:**

This paper proposes NI Sampling, a learnable framework to accelerate dLLMs by optimizing the token sampling order.

Instead of relying on heuristic confidence-based thresholds, the method introduces a neural indicator that predicts which masked tokens can be safely revealed at each step. To train this indicator, the authors design a trajectory-preserving objective, ensuring consistency with reference sampling trajectories while achieving significant speedups.

Empirical results show up to 14× acceleration compared to full-step sampling with negligible performance drop across benchmarks such as GSM8K, MATH, HumanEval, and MBPP.

**Strengths:**

* **Learnable Scheduler**
  Improving the sampling efficiency of discrete diffusion models is an important and timely research direction. The idea of approaching this problem through a *learnable* mechanism for sampling order optimization is particularly interesting and potentially impactful.

**Weaknesses:**

* **Overfitting to Reference Trajectory**
  (Please correct me if I have misunderstood this point.)
  A major limitation of NI Sampling is that it appears to produce generations that overfit to the *reference trajectory* used during training. In this sense, it differs fundamentally from efficient sampling studies in *continuous* diffusion models, where speedup is achieved by reducing *numerical errors* of ODE solvers rather than altering the learned data distribution.

While the paper claims acceleration in sampling speed, this efficiency primarily stems from treating tokens as *conditionally independent*—i.e., ignoring inter-token correlations. In contrast, methods that preserve token correlations (e.g., [1]) achieve efficiency without such independence assumptions. Consequently, NI Sampling resembles trajectory-conditioned approaches such as [2], which restrict the model’s output distribution to a small subset of trajectories, potentially degrading sample diversity.

[1]: *Jump Your Steps: Optimizing Sampling Schedule of Discrete Diffusion Models*, [https://openreview.net/forum?id=pD6TiCpyDR](https://openreview.net/forum?id=pD6TiCpyDR)
[2]: *Beyond Autoregression: Fast LLMs via Self-Distillation Through Time*, [https://arxiv.org/abs/2410.21035](https://arxiv.org/abs/2410.21035)

**Questions:**

* **Diversity and Trade-off Analysis**
  As mentioned above, the main concern is that NI Sampling might overfit to the reference trajectory, thus harming sample diversity. To quantify this, please consider reporting:

  1. **Token entropy**, to measure the diversity of generated tokens.
  2. **pass@K**, to evaluate the diversity–accuracy trade-off more explicitly.

Additionally, the paper would benefit from a **theoretical analysis** of when and why merging steps (as done in the trajectory-preserving principle) does not harm the underlying distribution. In other words, even if the merged distribution deviates from the base model’s, a formal understanding of this trade-off would allow users to make informed decisions about when NI Sampling is appropriate. Currently, the method lacks such theoretical grounding, making it unclear under what conditions the distribution shift is acceptable.

[1]: *Generative Flows on Discrete State-Spaces: Enabling Multimodal Flows with Applications to Protein Co-Design*, [https://arxiv.org/abs/2402.04997](https://arxiv.org/abs/2402.04997)

---

> ### Author Response · Authors · 2025-11-26
> **Response to Reviewer B3No 1**
>
> - Thank you for your constructive questions! We are pleased to see your recognition of optimizing sampling schedule as an interesting and timely research direction. We respond to your concerns below. We are happy to discuss further if you still have questions.
>
> > ### W1 & Q1: Overfitting to deterministic trajectory and lack of sampling diversity
>
>   - First, we would like to emphasize that greedy sampling is generally acceptable in many practical scenarios, such as solving math problems. In these cases, our proposed approach offers substantial practical value. Moreover, the greedy sampling setting is widely adopted as the default configuration in recent works on discrete large language models—for example, LLaDA (see https://github.com/ML-GSAI/LLaDA/blob/570f29032d6824ea14977c89a8eb402e6eb25f96/generate.py#L154) and Fast-dllm (see https://github.com/NVlabs/Fast-dLLM), which introduces confidence-threshold sampling and serves as our primary baseline. Following this line of work, we choose greedy sampling as our primary evaluation setting.
>
>   - **Nevertheless, ensuring sampling diversity and preventing overfitting to deterministic trajectories are crucial.** To address this concern, we provide additional experiments and discussion. Specifically, we first show that our current indicator does not degrade diversity. Then, by training a new indicator using random trajectories, we demonstrate that sampling diversity can be further improved. These results confirm that our method does not overfit to deterministic trajectories.
>
>   - Firstly, our current indicator can be incorporated with random sampling easily without additional tuning.
>     - Method: specifically, similar to the implementation of confidence threshold sampling, we sample token at each position randomly according to the probability vector instead of using argmax. Then, we feed all actually sampled tokens together with its probability into the existing neural indicator and let it make decisions.
>     - Further discussion:
>       - **Our method follows the same paradigm as the random sampling implementation of confidence threshold sampling.** Specifically, under the greedy setting, both methods deterministically sample all tokens and keep the tokens whose scores exceed a threshold. When combined with random sampling, the modification remains exactly the same for both methods: deterministic selection is simply replaced by sampling from the predicted distribution. The only difference between the two methods lies in how eligible tokens are determined—confidence-threshold sampling uses confidence values, while our method uses the indicator score.
>       - **In fact, our method may be even more diverse than confidence threshold sampling.** At each step, the set of tokens that can be accepted under greedy sampling is larger in our method than in confidence-threshold sampling. Consequently, under random sampling, there is a higher probability of selecting tokens that deviate from the original trajectory. This increases the likelihood of trajectory divergence and thus leads to greater sampling diversity.
>
>   - The diversity of our method can be further enhanced by modifying the training process.
>     - Method: since our existing indicator has only seen tokens sampled via greedy sampling and deterministic trajectories, we revised the training strategy and retrained the model as follows:
>       - The deterministic trajectory generation process is replaced with a stochastic process using a temperature of 1.0. **This addresses the reviewer’s concern regarding potential overfitting to deterministic trajectories.**
>       - During training, tokens are sampled according to the model’s predicted probabilities rather than being selected solely with greedy strategy, and these sampled tokens serve as inputs to the indicator.

---

> ### Author Response · Authors · 2025-11-26
> **Response to Reviewer B3No 2**
>
> (continued from previous response)
>
> - **Experiments.** We present experimental results of diversity evaluation to show that our existing neural indicator is fully compatible with stochastic sampling.
>     - **Setting and Evaluation.** We use LLaDA-8B-Instruct as the pre-trained model and evaluate on GSM8K and HumanEval. In addition, we include SQuAD Completion, SQuADv2, and TruthfulQA to cover a broader range of task types. We set the temperature to 1.0 for all experiments. For each problem, we generate $k$ different answers using multiple random seeds, and report the following metrics to quantify diversity:
>        - pass@k: A problem is considered correctly solved if at least one of the $k$ generated answers is correct. A diverse generative distribution should exhibit a steady increase in pass@k as $k$ grows. Note that we apply this metric to all datasets, not only code-related ones. For SQuADv2, whose official metric is the average F1 score, we compute the maximum F1 score among the $k$ independently generated answers for each problem and then report the average across all problems. We report pass@k under different values of $k$ and further compute the slope of the fitted line of pass@k with respect to $\log k$, which serves as our **primary metrics for evaluating diversity**. This metric is included following the suggestion of reviewer B3No.
>        - Self-BLEU [1]: For each answer, we compute its BLEU score with respect to the other $k-1$ answers. Then we take an average across all answers. We use both 1-gram-BLEU and 2-gram-BLEU. Lower values indicate higher diversity.
>        - Self-MAUVE [2]: We randomly split the $k$ answers into two groups and compute the MAUVE score between them. Lower values indicate higher diversity.
>
>         [1] Zhu et al., Texygen: A Benchmarking Platform for Text Generation Models.
>
>         [2] Pillutla et al., MAUVE: Measuring the Gap Between Neural Text and Human Text using Divergence Frontiers.
>
> - **Key Conclusions**
>
>   - For our existing indicator, experimental results are reported in the “old indicator” row of the following tables. The key conclusion is that **our existing indicator achieves comparable diversity to threshold sampling and full-step sampling across most datasets.** In most cases, the slope of pass@k with respect to $\log k$ is slightly higher than that of confidence threshold sampling, and similar to full-step sampling, indicating that our method achieves comparable diversity. Additionally, in most cases, our Self-BLEU and Self-MAUVE scores are comparable to or only slightly worse than those of confidence threshold sampling, further supporting the conclusion that the diversity of NI sampling is similar to baseline methods.
>
>   - For the indicator trained with random trajectories, experimental results are shown in the “new indicator” row of the following tables. **The key takeaway is that such indicator shows a larger slope of pass@k with respect to $\log k$, which indicates that training predictor on random trajectories can boost our sampling diversity.**
>
> - Overall, our method fully supports random sampling, and experimentally achieves performance comparable to full-step sampling, without overfitting to any specific deterministic trajectory. **This indicates that our approach is fully practical.**
>
> - We have updated the relevant content in Section C in the appendix of our revised paper.
>
> - Results are shown below.
>
>     - HumanEval-256 dataset
>
>     |               | Token/s | NFE   | Pass@1 | Pass@2 | Pass@4 | Pass@8 | Pass@16 | **Acc-log k slope$\uparrow$** |
>     |---------------|---------|-------|--------|--------|--------|--------|---------|---------------------------------|
>     | Full-step     | 42.91   | 256   | 36.99  | 44.69  | 51.43  | 57.81  | 63.76   | 6.67                            |
>     | Threshold     | 110.82  | 99.13 | 36.82  | 44.35  | 51.02  | 57.40  | 63.10   | 6.56                            |
>     | Old Indicator | 164.09  | 63.34 | 36.43  | 44.13  | 51.09  | 57.63  | 63.62   | 6.79                            |
>     | New Indicator | 155.78  | 66.72 | 36.70  | 44.38  | 51.34  | 58.03  | 64.33   | **6.89**                        |

---

> ### Author Response · Authors · 2025-11-26
> **Response to Reviewer B3No 3**
>
> (continued from previous response)
>
> - Results are shown below.
>
>     - GSM8K-256 dataset
>
>     |               | Token/s | NFE   | Pass@1 | Pass@2 | Pass@4 | Pass@8 | **Acc-log k slope$\uparrow$** | Self-1-gram-BLEU$\downarrow$ | Self-2-gram-BLEU$\downarrow$ | Self-MAUVE$\downarrow$ |
>     |---------------|---------|-------|--------|--------|--------|--------|---------------------------------|--------------------------------|--------------------------------|--------------------------|
>     | Full-step     | 19.11   | 256   | 76.95  | 84.83  | 89.99  | 93.48  | 5.47                            | 0.916                          | 0.883                              | 0.966                    |
>     | Threshold     | 65.56   | 74.63 | 77.55  | 85.51  | 91.05  | 93.85  | 5.44                            | 0.920                          | 0.887                              | 0.974                    |
>     | Old Indicator | 86.74   | 54.70 | 77.78  | 85.97  | 91.05  | 93.70  | 5.28                            | 0.919                          | 0.886                              | 0.971                    |
>     | New Indicator | 82.36   | 57.68 | 77.33  | 84.91  | 90.83  | 94.47  | **5.73**                        | 0.918                          | 0.884                          | 0.973                    |
>
>     - Squad Completion
>
>     |               | Token/s | NFE   | pass@1 | pass@2 | pass@4 | pass@8 | **Acc-log k slope$\uparrow$** | self-1-gram-BLEU$\downarrow$ | self-2-gram-BLEU$\downarrow$ | Self-MAUVE$\downarrow$ |
>     |---------------|---------|-------|--------|--------|--------|--------|---------------------------------|--------------------------------|--------------------------------|--------------------------|
>     | full-step     | 48.48   | 128   | 78.3   | 83.3   | 88.2   | 91.3   | 4.39                            | 0.809                          | 0.600                          | 0.981                    |
>     | Threshold     | 174.64  | 34.03 | 77.5   | 84.0   | 87.0   | 89.5   | 3.90                            | 0.802                          | 0.600                          | 0.986                    |
>     | Old Indicator | 266.78  | 21.39 | 76.8   | 83.2   | 86.8   | 89.7   | 4.23                            | 0.810                          | 0.592                          | 0.982                    |
>     | New Indicator | 254.98  | 22.38 | 76.8   | 83.2   | 87.3   | 89.8   | **4.31**                        | 0.811                          | 0.589                          | 0.981                    |
>
>   - Squad V2
>
>    |               | Token/s | NFE   | pass@1 | pass@2 | pass@4 | pass@8 | **Acc-log k slope$\uparrow$** | Self-1-gram-BLEU$\downarrow$ | Self-2-gram-BLEU$\downarrow$ | Self-MAUVE$\downarrow$ |
>   |---------------|---------|-------|--------|--------|--------|--------|---------------------------------|--------------------------------|--------------------------------|--------------------------|
>     | full-step     | 47.94   | 128   | 27.64  | 31.76  | 37.46  | 41.92  | 4.85                            | 0.792                          | 0.710                          | 0.989                    |
>     | Threshold     | 202.27  | 31.23 | 28.05  | 33.21  | 37.92  | 42.21  | 4.72                            | 0.792                          | 0.710                          | 0.989                    |
>     | Old Indicator | 267.74  | 22.56 | 27.85  | 33.13  | 38.01  | 43.20  | 5.09                            | 0.797                          | 0.723                          | 0.989                    |
>     | New Indicator | 283.18  | 21.32 | 27.09  | 34.10  | 39.21  | 45.20  | **5.94**                        | 0.802                          | 0.720                          | 0.990                    |
>
>   - Truthful QA
>
>   |               | Token/s | NFE   | bleu_acc | rouge1_acc | rouge2_acc | rougeL_acc | Self-1-gram-BLEU$\downarrow$ | Self-2-gram-BLEU$\downarrow$ | Self-MAUVE$\downarrow$ |
>   |---------------|---------|-------|----------|------------|------------|------------|--------------------------------|--------------------------------|--------------------------|
>   | Full-step     | 46.54   | 128   | 53.86    | 53.61      | 49.07      | 52.88      | 0.898                          | 0.862                          | 0.983                    |
>   | Threshold     | 218.80  | 26.46 | 55.32    | 54.35      | 49.20      | 54.10      | 0.903                          | 0.867                          | 0.984                    |
>   | Old Indicator | 419.67  | 13.41 | 57.89    | 56.55      | 52.39      | 57.04      | 0.903                          | 0.867                          | 0.980                    |
>   | New Indicator | 408.94  | 13.90 | 61.57    | 59.00      | 54.59      | 59.73      | 0.901                          | 0.864                          | 0.985                    |

---

> ### Author Response · Authors · 2025-11-26
> **Response to Reviewer B3No 4**
>
> (continued from previous response)
>
> - Other discussions regarding the review
>
>    - **Diversity evaluation metrics.** We sincerely appreciate your suggestion to use pass@k and token entropy. We adopt pass@k as the primary metric for evaluating diversity. Regarding token entropy, we note that computing average token entropy is less meaningful in discrete diffusion models because multiple tokens are sampled at each step, and the sampled set inevitably includes tokens with low confidence, inflating the entropy. Moreover, for each token, the predicted probability does not directly reflect the actual probability of being finally sampled, as the remasking process also influences the outcome. These factors make entropy difficult to estimate reliably. Therefore, we do not report token entropy and instead use metrics such as Self BLEU and Self-MAUVE to measure correlations among the generated answers.
>
>   - **Discussion on related works mentioned by the reviewer**
>     - Jump Your Steps [1], mentioned by the reviewer, explicitly optimizes the upper bound of the distribution loss. However, it uses a uniform timestep schedule for all prompts, which may lead to distribution loss (please correct us if we misunderstand this work), as the optimal schedule may vary across inputs. In contrast, while our method does not explicitly optimize distribution loss, it makes sample-dependent decisions and therefore does not necessarily underperform in terms of distribution preservation.
>     - **Discussion on the difference of sampling optimization between discrete and continuous diffusion models.** The sampling process in continuous diffusion models can be viewed as numerically solving an ODE, which makes minimizing numerical error important. In contrast, discrete diffusion models operate differently: sampling proceeds by selecting a discrete set of token positions at each step rather than following a continuous ODE trajectory. As a result, the analytical tools used for continuous diffusion are not directly applicable. For discrete diffusion, the most direct way to optimize sampling is to decide **which tokens to sample**, which is precisely the focus of this work.
>
>   [1]: Jump Your Steps: Optimizing Sampling Schedule of Discrete Diffusion Models
>
> > ### Q2: theoretical analysis of when and why merging steps does not harm the underlying distribution
>
>   - **For the greedy sampling case**, as discussed in Sec. 3.2, merging adjacent steps along the trajectory does not affect the final sampling result. Specifically, for the original sampling trajectory, we merge the subsequent step into the current step only when the current model prediction at all positions in the subsequent step aligns with the final results. In this way, sampling these two steps simultaneously incurs no impact on later steps. Therefore, merging steps will never harm the sampling results of the original trajectory.
>
>   - If the reviewer means the distribution of the random sampling case, we firstly would like to point out that, without retraining the base model, **it is almost impossible to simultaneously sample multiple tokens per step and avoid any distribution loss with random sampling**. This would require that, (1) at every sampling step, there are multiple masked tokens that are completely conditionally independent of each other, which is an assumption that rarely holds in practice, and (ii) that there exists a mechanism capable of precisely identifying such independent tokens without wasting NFEs. Moreover, inter-token dependency can vary significantly across different prompts. Considering all these factors, guaranteeing no distribution loss is extremely difficult. This issue holds for every method that tries to sample multiple tokens within one step, including confidence threshold sampling and our method. **Given the fact that the accuracy and diversity of our method is comparable to full-step sampling and confidence threshold sampling**, we believe that it is not necessary to guarantee completely eliminating distribution loss.
>
>   - To fully address the reviewer's concern, we further provide theoretical analysis regarding the distribution loss of merging two adjacent steps along the trajectory in the next part of our rebuttal

---

> ### Author Response · Authors · 2025-11-26
> **Response to Reviewer B3No 5**
>
> (continued from previous response)
>
> - In this analysis, we mainly follow the discussion in the paper of confidence threshold sampling [1]. We assume the probability given by the pre-trained discrete diffusion model is accurate.
>
>     - **Problem definition and notations.** Since our training process uses confidence threshold sampling as the base sampling method to generate our reference trajectory, we analyze step merging under this setting. Under the greedy setting, assuming the threshold of the base sampling method is $1-\epsilon$, the reference trajectory decodes $n_1$ tokens at the current step, and will decode $n_2$ tokens at the next step. We denote the random variable of the first group as $X_1$ and greedy sampling result $x_1$, and the random variable of the second group as $X_2$ and greedy sampling result $x_2$. Now our NI sampling will decode $X_1$ and $X_2$ simultaneously at the current step. We denote the minimum confidence of $X_2$ at the current step as $1-\epsilon^{\prime}$. We denote $p(X_1, X_2)$ as the distribution of sampling $X_1$ and $X_2$ in two separate steps, respectively; $q(X_1, X_2)$ as the distribution of sampling $X_1$ and $X_2$ in a single merged step. We calculate $L_p$ distance between these two distributions as the metric to evaluate the distribution gap.
>
>     - We first give the bound of $|p((X_1,X_2)=(x_1,x_2))-q((X_1,X_2)=(x_1,x_2))|$
>
>       - $p((X_1,X_2)=(x_1,x_2))=p(X_1=x_1)p(X_2=x_2|X_1=x_1)$. Consider $p(X_1=x_1)$, which can be given as $P(X_1=x_1)=\prod_{j=1}^{n_1}(1-\epsilon_j)$, where $1-\epsilon_j$ is the confidence of the j-th token. The bound of $P(X_1=x_1)$ can be given as:
>
>            $1-n_1\epsilon\leq1-\sum_{j=1}^{n_1}\epsilon_j\leq P(X_1=x_1)\leq 1-\epsilon$
>
>       - Similarly, the bound of $P(X_2=x_2|X_1=x_1)$ can be given as:
>
>            $1-n_2\epsilon\leq1-\sum_{j=1}^{n_2}\epsilon_j\leq P(X_2=x_2|X_1=x_1)\leq 1-\epsilon$
>
>       - So the bound of $p((X_1,X_2)=(x_1,x_2))$ is:
>
>            $(1-n_1\epsilon)(1-n_2\epsilon)\leq p((X_1,X_2)=(x_1,x_2))\leq (1-\epsilon)^2$
>
>     - Then we calculate the bound of $q((X_1,X_2)=(x_1,x_2))$. In a similar way, the bound can be given as:
>
>          $1-n_1\epsilon-n_2\epsilon^{\prime}\leq q((X_1,X_2)=(x_1,x_2))\leq 1-\epsilon^{\prime}$
>
>     - So $|p((X_1,X_2)=(x_1,x_2))-q((X_1,X_2)=(x_1,x_2))|\leq max(|1-\epsilon^{\prime}-(1-n_1\epsilon)(1-n_2\epsilon)|,|(1-\epsilon)^2+n_1\epsilon+n_2\epsilon^{\prime}-1|)$
>
>     - For $\sum_{(z_1,z_2)\neq(x_1,x_2)}|p(z_1,z_2)-q(z_1,z_2)|$, its bound can be given as:
>
>          $\sum_{(z_1,z_2)\neq(x_1,x_2)}|p(z_1,z_2)-q(z_1,z_2)|\leq p((X_1,X_2)\neq(x_1,x_2))+q((X_1,X_2)\neq(x_1,x_2))\leq 2n_1\epsilon+n_2(\epsilon^{\prime}+\epsilon)-n_1n_2\epsilon^2$
>
>     - We can easily get that for all $(z_1,z_2)\neq(x_1,x_2)$, $|p(z_1,z_2)-q(z_1,z_2)|\leq\epsilon$
>
>     - Thus, $\sum_{(z_1,z_2)\neq(x_1,x_2)}|p(z_1,z_2)-q(z_1,z_2)|^p\leq(max_{(z_1,z_2)\neq(x_1,x_2)}|p(z_1,z_2)-q(z_1,z_2)|)^{p-1}\sum_{(z_1,z_2)\neq(x_1,x_2)}|p(z_1,z_2)-q(z_1,z_2)|\leq\epsilon^{p-1}(2n_1\epsilon+n_2(\epsilon^{\prime}+\epsilon)-n_1n_2\epsilon^2)$
>
>     - Now we can give the bound of $L_p$ distance: $$L_p(p,q)=(|p((X_1,X_2)=(x_1,x_2))-q((X_1,X_2)=(x_1,x_2))|^p+\sum_{(z_1,z_2)\neq(x_1,x_2)}|p(z_1,z_2)-q(z_1,z_2)|^p)^{1/p}\leq (max(|1-\epsilon^{\prime}-(1-n_1\epsilon)(1-n_2\epsilon)|^p,|(1-\epsilon)^2+n_1\epsilon+n_2\epsilon^{\prime}-1|^p)+\epsilon^{p-1}(2n_1\epsilon+n_2(\epsilon^{\prime}+\epsilon)-n_1n_2\epsilon^2))^{1/p}$$
>
>     - Since all tokens in $X_2$ should not be sampled in the first step, so $\epsilon^{\prime}>\epsilon$; Besides, since $X_2$ should be sampled in the second step, so $1-\epsilon^{\prime}\geq(1-\epsilon)(1-n_1\epsilon)$.  We can further substitute this into the above equation to eliminate $\epsilon^{\prime}$: $$L_p(p,q)\leq (max(|1-\epsilon-(1-n_1\epsilon)(1-n_2\epsilon)|^p,|-\epsilon^2(n_2n_1-1)+(n_1n_2+n_1+n_2-2)\epsilon|^p)+\epsilon^{p-1}((2(n_1+n_2)+n_1n_2)\epsilon-2n_1n_2\epsilon^2))^{1/p}=(|-\epsilon^2(n_2n_1-1)+(n_1n_2+n_1+n_2-2)\epsilon|^p+\epsilon^{p-1}((2(n_1+n_2)+n_1n_2)\epsilon-2n_1n_2\epsilon^2))^{1/p}$$
>
> [1] Wu et al., Fast-dLLM: Training-free Acceleration of Diffusion LLM by Enabling KV Cache and Parallel Decoding

---

### Official Review · Reviewer_VVhC · 2025-11-01

**Soundness:** 3
**Presentation:** 3
**Contribution:** 3
**Rating:** 8
**Confidence:** 3

**Summary:**

This paper proposes NI Sampling, an efficient decoding framework for pre-trained diffusion LLMs that optimizes token sampling order with a tiny neural indicator predicting which masked positions can be simultaneously and safely revealed at each step. Labels for the indicator are derived from a trajectory-preserving criterion that merges steps when the current predictions already match a reference trajectory, turning correct predictions into parallel. On LLaDA-8B/1.5 and Dream-7B across GSM8K, MATH, MBPP, and HumanEval, NI Sampling delivers significant speedups over full-step sampling and threshold-based sampling with negligible accuracy loss.

**Strengths:**

- I enjoyed reading this paper. It's well-motivated. The trajectory-preserving case is great and clear, demonstrating the great potential of merging sampling steps for acceleration
- The proposed solution is conceptually simple and practical. A lightweight MLP as the indicator brings marginal extra cost.
- The proposed method achieves consistent and significant speedups across multiple benchmarks compared to threshold-based sampling and full-step sampling, with negligible accuracy loss.
- Detailed description of the setup of NI, analysis, and ablation studies, in addition to the main results.

**Weaknesses:**

I think this paper and its contributions could be further enhanced with more investigation into the transferability of the neural indicator:
- I got the difference between LLaDA models and Dream, but for models from the same family (e.g., LLaDA-8B and LLaDA-1.5 used in this paper), would it be possible to use a shared pretrained neural indicator?
- Would it be possible to use the same pretrained neural indicator for different generation window lengths?

**Questions:**

Please see Weaknesses

---

> ### Author Response · Authors · 2025-11-26
> **Response to Reviewer VVhC 1**
>
> - We sincerely thank your positive feedback and thoughtful questions! We are pleased to see your acknowledgement of the potential, novelty, and effectiveness of our method. We respond to your concerns below. We are happy to discuss further if you still have questions.
>
> > ### W1.1: Would it be possible to use a shared pretrained neural indicator?
>   - In response to your question, we evaluate the indicators trained on LLaDA-8B-Instruct and LLaDA-1.5 by testing them on LLaDA-1.5 and LLaDA-8B-Instruct, respectively. Results are shown in tables below. The resulting Acc–NFE pairs are reported in the “Swapped Indicator” row. For comparison, we also include results for the confidence-threshold method and the original indicator in the “Baseline” and “Original Indicator” rows, respectively.
>   - We can see that swapping the indicators trained on the two models does not result in a significant drop in performance. This suggests that our method exhibits a certain degree of generalization within the same model family. We propose several potential reasons for this phenomenon:
>     - LLaDA-1.5 was fine-tuned from LLaDA-8B-Instruct, so their hidden states are naturally quite similar.
>     - Our indicator uses the confidence information of sampled tokens as input, which provides a strong signal for decision-making and can be directly transferred across different models.
>
> - Results
>     - Results on GSM8K-128
>         - LLaDA-1.5 model + indicator trained with LLaDA-8B-Instruct
>
>         |     |         |      |        |       |       |       |       |    |
>         |--------------------|-----|-------|-------|-------|--------|--------|-------|-------|
>         |  Swapped Indicator |       NFE       | 15.16 | 16.70 | 18.41 | 20.32 | 23.25 | 26.81 | 34.33 |
>         |  Swapped Indicator |       Acc       | 62.17 | 67.63 | 69.37 | 71.04 | 72.86 | 75.13 | 75.28 |
>         | Original Indicator |       NFE       | 13.33 | 15.27 | 17.18 | 18.96 | 21.71 | 26.11 | 29.89 |
>         | Original Indicator |       Acc       | 57.01 | 62.47 | 68.61 | 71.19 | 73.46 | 74.37 | 76.19 |
>         |      Baseline      |       NFE       | 11.91 | 14.60 | 17.64 | 21.60 | 27.38 | 35.63 | 47.44 |
>         |      Baseline      |       Acc       | 37.00 | 42.53 | 51.86 | 59.97 | 68.16 | 75.44 | 75.82 |
>
>         - LLaDA-8B-Instruct model + indicator trained with LLaDA-1.5
>
>         |     |         |      |        |       |       |       |       |    |
>         |--------------------|-----|-------|-------|-------|--------|--------|-------|-------|
>         |  Swapped Indicator |       NFE       | 13.72 | 15.30 | 17.10 | 19.22 | 21.95 | 25.71 | 33.22 |
>         |                    |       Acc       | 54.06 | 61.79 | 66.49 | 67.25 | 71.65 | 72.25 | 74.22 |
>         | Original Indicator |       NFE       | 15.47 | 16.80 | 18.65 | 20.84 | 23.51 | 27.41 | 32.73 |
>         |                    |       Acc       | 60.35 | 64.44 | 67.02 | 69.37 | 70.89 | 71.95 | 73.84 |
>         |      Baseline      |       NFE       | 11.88 | 14.85 | 17.97 | 22.16 | 28.11 | 37.15 | 49.35 |
>         |                    |       Acc       | 35.25 | 43.06 | 52.08 | 59.74 | 68.92 | 73.01 | 73.77 |
>      - Results on MATH-128
>          - LLaDA-1.5 model + indicator trained with LLaDA-8B-Instruct
>
>         |     |         |      |        |       |       |       |       |    |
>         |--------------------|-----|-------|-------|-------|--------|--------|-------|-------|
>         |   Swapped Indicator |       NFE       | 15.17 | 17.13 | 19.36 | 22.16 | 25.64 | 30.29 | 41.22 |
>         |   Swapped Indicator |       Acc       | 22.35 | 23.13 | 25.18 | 26.65 | 28.28 | 28.82 | 30.54 |
>         | Original Indicator |       NFE       | 15.89 | 18.37 | 20.59 | 23.36 | 26.54 | 30.49 | 38.16 |
>         | Original Indicator |       Acc       | 21.52 | 24.29 | 25.62 | 27.19 | 28.83 | 29.72 | 30.51 |
>         |      Baseline      |       NFE       | 20.57 | 24.12 | 28.20 | 33.24 | 39.52 | 47.28 | 58.27 |
>         |      Baseline      |       Acc       | 17.14 | 20.06 | 23.59 | 26.32 | 28.78 | 30.09 | 30.93 |
>
>           - LLaDA-8B-Instruct model + indicator trained with LLaDA-1.5
>
>         |     |         |      |        |       |       |       |       |    |
>         |--------------------|-----|-------|-------|-------|--------|--------|-------|-------|
>         |  Swapped Indicator |       NFE       | 16.89 | 19.32 | 21.85 | 24.57 | 27.79 | 31.94 | 37.57 |
>         |  Swapped Indicator  |       Acc       | 19.77 | 22.89 | 24.04 | 26.22 | 27.63 | 28.53 | 29.10 |
>         | Original Indicator |       NFE       | 15.88 | 18.16 | 20.08 | 23.58 | 27.47 | 32.35 | 39.96 |
>         | Original Indicator |       Acc       | 22.11 | 23.32 | 25.32 | 27.13 | 28.10 | 29.21 | 29.90 |
>         |      Baseline      |       NFE       | 20.85 | 24.57 | 28.99 | 34.30 | 40.97 | 49.19 | 60.42 |
>         |      Baseline  |       Acc       | 15.44 | 19.43 | 23.39 | 26.23 | 27.68 | 29.05 | 29.54 |

---

> ### Author Response · Authors · 2025-11-26
> **Response to Reviewer VVhC 2**
>
> (continued from previous response)
>
> - Results
>
>     - Results on HumanEval-256
>
>         - LLaDA-1.5 model + indicator trained with LLaDA-8B-Instruct
>
>         |                    |     |       |       |       |       |       |       |        |
>         |--------------------|-----|-------|-------|-------|-------|-------|-------|--------|
>         |  Swapped Indicator | NFE | 21.83 | 26.87 | 30.69 | 36.38 | 42.25 | 50.25 |  66.67 |
>         |  Swapped Indicator | Acc | 10.37 | 17.68 | 26.22 | 25.61 | 37.80 | 38.41 |  41.46 |
>         | Original Indicator | NFE | 22.78 | 27.62 | 31.01 | 37.56 | 43.65 | 51.71 |  64.40 |
>         | Original Indicator | Acc | 11.59 | 20.73 | 23.78 | 29.88 | 33.54 | 36.59 |  42.68 |
>         |      Baseline      | NFE | 26.01 | 32.65 | 40.50 | 49.65 | 60.12 | 75.14 | 100.71 |
>         |      Baseline      | Acc |  6.10 | 15.24 | 25.00 | 31.71 | 40.24 | 40.24 |  42.68 |
>
>         - LLaDA-8B-Instruct model + indicator trained with LLaDA-1.5
>
>         |                    |     |       |       |       |        |        |       |       |
>         |--------------------|-----|-------|-------|-------|--------|--------|-------|-------|
>         |  Swapped Indicator | NFE | 19.69 | 23.71 | 28.15 | 32.25  | 40.34  | 48.98 | 69.46 |
>         |  Swapped Indicator | Acc | 11.59 | 17.68 | 27.44 | 25.61  | 32.32  | 37.80 | 38.41 |
>         | Original Indicator | NFE | 24.64 | 28.60 | 33.98 | 40.79  | 52.53  | 58.25 | 68.07 |
>         | Original Indicator | Acc | 21.95 | 28.66 | 31.71 | 33.54  | 35.98  | 37.20 | 39.02 |
>         |      Baseline      | NFE | 22.12 | 29.06 | 37.74 | 49.12  | 58.15  | 75.14 | 98.65 |
>         |      Baseline      | Acc | 7.93  | 17.07 | 23.17 | 28.66  | 34.15  | 36.60 | 37.20 |
>
> > ### W1.2: Would it be possible to use the same pretrained neural indicator for different generation window lengths?
>   - In our current experiments across different generation lengths, we already use the same indicator. Specifically, for each pre-trained dLLM, we train a single, general indicator that is used for all tasks and generation lengths.

---

### Author Response · Authors · 2025-11-26
**General Response**

We sincerely thank all the reviewers for their valuable and constructive feedback. We are pleased that reviewers acknowledge our **strong motivation**, **significant speedup ratio achieved by NI sampling**, and **clear presentation**. We also appreciate the recognition that **sampling order optimization is particularly interesting and shows great acceleration potential**, and that **leveraging a lightweight neural indicator is a simple yet effective design choice**. Below, we summarize the main updates in our rebuttal. Detailed discussions are provided in the responses to individual reviewers.
  - **We show that our current method already naturally supports random sampling.** Concretely, instead of taking the argmax, we sample each token according to its probability distribution, then feed the sampled tokens to the neural indicator for decision making.
  - **We conduct additional experiments to evaluate the sampling diversity of our method.** We include diversity evaluation metrics such as pass@k, self-BLEU and self-MAUVE. Besides, we also add datasets beyond math and code generation to provide a more comprehensive evaluation.
  - **We further train a new indicator with randomly sampled trajectories.** This prevents indicator from overfitting on deterministic trajectories. We show that this training strategy further improves the sampling diversity of NI sampling.
  - **We analyze why the practical speedup does not fully reach the theoretical upper bound presented in Section 3.** Specifically, we show several practical cases where the indicator struggles to learn the ground-truth labels and identify the issues.

---

### Author Response · Authors · 2025-12-03
**Summary of Paper and Discussion Part (1/2)**

Dear Reviewers and ACs,
  - We sincerely thank you for your time and effort in reviewing our paper. Below, we provide a concise summary of our work, our responses to reviewers' concerns, and an overview of the revisions we made.

  ---

  - ### Summary of Our Paper

    Our paper focuses on reducing the number of sampling steps required by discrete diffusion models.

    We observe that, in discrete diffusion models, many tokens are predicted correctly at each step, but most of them cannot be sampled due to low confidence. To demonstrate that discrete diffusion can be substantially accelerated by selecting the appropriate positions to unmask for each step (i.e., token order), we propose a **trajectory-preserving order**. At every step, we sample all tokens that are predicted correctly and follow the correct trajectory order (Figure 2). This can also be viewed as merging adjacent steps whenever the corresponding predicted tokens at the first step align with the final results (Algorithms 1 and 2).  Our results show that, with proper selection of sampling positions, the maximum achievable acceleration can reach up to **24$\times$**.

    Building on this, we propose **NI Sampling**, which uses a lightweight neural indicator to decide which positions should be unmasked at each step. We first generate a large number of reference trajectories. At every step, we label whether each token should be sampled according to the trajectory-preserving order. These labels are then used to train the indicator. During inference, the indicator assigns a score to each masked token to determine if it should be sampled. Our indicator generalizes well across datasets and generation lengths. Our method achieves up to 14.3$\times$ inference acceleration while completely dominating the accuracy–step curves of confidence-threshold baselines.
  ---
  - ### Strengths Recognized by Reviewers
    - We are pleased that reviewers acknowledge our key contributions, including strong motivation and potential of token order optimization, our straightforward and effective method, substantial improvements in experimental results, and clear presentation:
    - **Strong potential and interesting insights of token order optimization**
      - **Reviewer VVhC**: "It's well-motivated. The trajectory-preserving case is great and clear, demonstrating the great potential of merging sampling steps for acceleration."
      - **Reviewer B3No**: "Learnable Scheduler Improving the sampling efficiency of discrete diffusion models is an important and timely research direction. The idea of approaching this problem through a learnable mechanism for sampling order optimization is particularly interesting and potentially impactful."
      - **Reviewer wmZ5**: "The idea of merging multiple sampling steps into a single step is simple yet presents an interesting direction for accelerating generation."
      - **Reviewer Duu3**: "This paper offers an evidence that the optimization of sampling positions in masked diffusion can greatly accelerate the sampling process, even without changing the output/trajectory...This offers a deeper understanding of masked diffusion sampling from a novel angle."
    - **Straightforward and effective method**
      - **Reviewer VVhC**: "The proposed solution is conceptually simple and practical."
      - **Reviewer Duu3**: "The neural indicator learning also has a straightforward learning signal."
    - **Strong experimental results**
      - **Reviewer VVhC**: "The proposed method achieves consistent and significant speedups across multiple benchmarks compared to threshold-based sampling and full-step sampling, with negligible accuracy loss."
      - **Reviewer wmZ5**: The proposed merging strategy significantly speeds up sampling while maintaining performance.
      - **Reviewer Duu3**: "The neural indicator succeeds in actually accelerating masked diffusions."
    - **Clear writing and presentation**
      - **Reviewer VVhC**: "I enjoyed reading this paper." "Detailed description of the setup of NI, analysis, and ablation studies, in addition to the main results."
      - **Reviewer Duu3**: "This paper is well written."
  ---

---

> ### Author Response · Authors · 2025-12-03
> **Summary of Paper and Discussion Part (2/2)**
>
> (continued from previous comment)
>   - ### Primary Concerns Raised by Reviewers and Our Responses
>     - During rebuttal, we provided detailed discussions and added extensive experiments to address reviewers' concerns. Our responses to main issues are summarized below.
>     - Whether our method overfits to deterministic trajectories and reduces the diversity of sampling (B3No，wmZ5，Duu3).
>       - Firstly, we point out that many existing works use greedy sampling as their default setting, like LLaDA full-step implementation and confidence threshold sampling.
>       - Although we use deterministic sampling in our paper, our method can easily ensure randomness. Similar to confidence threshold sampling, we just replace argmax selection with probability-based sampling at each position, and then use the neural indicator to make decisions for the actually sampled tokens.
>         - We further extend our evaluation with the **pass@k** metric following **reviewer B3No**, and use the slope of the fitted line of accuracy v.s. $\log k$ as the primary diversity measure. We also report Self-BLEU and Self-MAUVE as additional diversity metrics and include three new datasets following **reviewer wmZ5** and **Duu3**. These experiments show that **the original indicator already achieves diversity comparable to full-step sampling and confidence-threshold methods**.
>       - To further address concerns, we trained a new neural indicator using randomly generated trajectories, where tokens at each position were also sampled randomly. **This new indicator exhibits higher diversity than the original indicator, showing that our method is not limited to deterministic trajectory during training**.
>       - We also provide theoretical derivations to quantify the distributional gap between our method and baselines.
>     - Why our method does not reach the upper bound in Section 3 (wmZ5，Duu3).
>       - First, we clarify that identifying the ideal upper bound and proposing NI Sampling are two independent contributions. NI Sampling provides a practical acceleration method, while the upper bound reveals potential and inspires future work.
>       - We supplement several examples demonstrating certain limitations of the indicator following **reviewer Duu3**, and suggest possible directions for future work.
>     - Generalization of the indicator across generation lengths and models (VVhC)
>       - In the original experiments, the same indicator was used for all generation lengths. We further include experiments showing that the indicator generalizes across different models within the same family.
>     - Comparison with additional baselines (wmZ5)
>       - We added comparisons and discussions of the suggested baselines.
>   ---
>   - ### Reviewer feedback
>     - **Reviewer wmZ5 thinks we have addressed most of his concerns and has raised his rating from 2 to 6.** He also provided two additional suggestions, which we have addressed in our follow-up comments.
>     - **Reviewer Duu3 states that our rebuttal has addressed all his concerns and has raised his rating from 6 to 8.**
>     - Although VVhC and B3No did not respond to our rebuttal during the discussion period, we believe our clarifications and additional experiments have also effectively addressed their concerns.
>   ---
>   - ### Major Update in Our Revision
>     - We include the discussion of the random sampling method and additional diversity evaluation experiments in Section C of the Appendix.
>     - We provide examples that illustrate certain limitations of the indicator in Section E of the Appendix.
>     - We add discussion of the newly suggested baseline methods in Section 6, with corresponding comparison results presented in Section D of the Appendix.
>     - We have also fixed typos.

---

### Meta-Review · Area_Chair_J1s1 · 2026-01-08

**Summary:**

This paper proposes neural indicator sampling (NSI), a method for speeding up dLLM sampling. The key idea is that one can train a neural network on a set of dLLM sampling trajectories to predict the token merging patter during sampling. With the help of NSI, the authors are able to achieve significant speedup with little performance drop.

The discussions boil down to two main concerns, the sample diversity from NSI and related how generalizable NSI is. The authors did an excellent job responding to each point with both explanations and substantial experiments. As a result, two reviewers have indicated increasing their ratings. Effectively, the AC believes that there is no major concerns that prevents this work from being accepted. Overall, it's a simple idea that addresses an important problem and shows promising results, the AC recommends accept.

**Reviewer Concerns:**

All major concerns are effectively addressed.

**Reviewer Scores:**

In the ideal world I can see all reviewers being willing to increase their ratings.

---

### Decision · Program_Chairs · 2026-01-26

Accept (Poster)